# Single cell profiling framework reveals metabolic subpopulations as drivers of bioproduction heterogeneity

Juline Savigny[1], Kiyan Shabestary[1], Maria Portela [1], Cinzia Klemm [1], Yvette Sum[1], Piotr Hapeta[2,3], Marko Storch [2,3], Christopher Rowlands [1] & Rodrigo Ledesma-Amaro [1,4,5] ✉

Heterogeneity within clonal cell populations remains a critical bottleneck within bioprocess engineering, notably by undermining bioproduction yields. Efforts to mitigate its impact have, however, been hampered by technological difficulties quantifying metabolism at the single-cell level. Here, we propose a framework based on single-cell biosensor analysis that enables robust characterisation of cell's metabolic states, leveraging it to detect and isolate isogeneic heterogeneity in response to environmental perturbations and within microbial cell factories. We identify acute and gradual glucose depletion to induce differentiation of metabolically distinct subpopulations and reveal these subpopulations to exhibit differential production capabilities, with lower intracellular pH subpopulations exhibiting enhanced product accumulation within violacein-producing strains but reduced yields within lycopene-producing strains. Lastly, we highlight galactose cultivation as a method to modulate subpopulation dynamics towards higher-producing lycopene phenotypes. Altogether, our research provides insights into subpopulation differentiation and establishes promising avenues for the engineering of more robust and higher-producing strains.

Isogenic, or phenotypic, heterogeneity refers to phenotypic cell-to-cell variations observed within genetically identical cell populations[1]. It is a widespread phenomenon which plays a key role in the fitness of many organisms - from promoting survival when faced with environmental perturbations through bet-hedging[2–7] to allowing for division of labour strategies to occur[8–10]. Within bioproduction and the artificial environments that are bioreactors, however, isogenic heterogeneity is primarily associated with the emergence of low-producing subpopulations which impair product yields[11–16] and likely promote strain instability[1,17,18]. To date, efforts to counter the occurrence of such low-producing subpopulations have predominantly relied on the use of feedback loops to actively select for high-producing cells[13,16,18]. While

this approach has proven effective in some cases, it poses several limitations. Most notably, the construction of such feedback loops requires orthogonal product-responsive transcription factors, which are only available for a few select pathways. These feedback loops will also compete with the native metabolism and engineered pathways for cellular resources and thus, their incurred cost and subsequent impact on bioproduction yield needs to be carefully evaluated. Instead, a promising approach involves studying the underlying causes of isogenic heterogeneity to inform the design of more robust and higher-producing strains.

To that end, a comprehensive overview of single cells' metabolic states is essential. Metabolites serve as dynamic indicators of cellular

[1]Department of Bioengineering, Imperial College London, London, UK. [2]London Biofoundry, Translation and Innovation Hub, Imperial College White City Campus, London, UK. [3]Department of Infectious Disease, Imperial College London, London, UK. [4]Bezos Centre for Sustainable Protein, Imperial College London, London, UK. [5]UKRI Engineering Biology Mission Hub on Microbial Food, Imperial College London, London, UK. ✉e-mail: r.ledesma-amaro@imperial.ac.uk

activity, responding promptly to environmental cues and engaging in complex bidirectional regulation with gene networks. Monitoring of metabolites can thus be used as a proxy for isogenic heterogeneity, enabling its detection in response to both environmental and cellular triggers, and providing valuable insight into its biological function. Quantification of metabolites at the single-cell level, however, is technically challenging due to the low sensitivity of existing metabolomics platforms. Indeed, while efforts to address this are underway, single-cell metabolomics remain in their infancy and are thus prohibitively costly, low throughput, and not readily accessible[19]. Instead, the most valuable tools we have at our disposal are metabolite-responsive genetically encoded fluorescent biosensors. These, in combination with single-cell detection methods such as flow cytometry and microscopy, enable for the relative concentrations of metabolites to not only be quantified at single-cell level but also assessed in real-time through transient fluctuations. For instance, the use of an L-valine biosensor within microfluidics chambers allowed Mustafi et al. to track the lineage of low-producing L-valine subpopulations in *C. glutamicum*[11], while a pH biosensor in combination with flow cytometry enabled Bagamery et al. to better understand the bet-hedging phenotypes which emerged in response to glucose starvation in *S. cerevisiae*[6]. Despite the numerous valuable biosensors present within the literature[20–23], few have attempted to standardise biosensor parts, focusing primarily on population-level characterisation[24] and thus leaving single-cell evaluation of biosensor behaviour largely unexplored. Part standardisation adds uniformity and robustness to construct design which is crucial for improving both their predictability and reproducibility. This is especially significant when studying isogenic heterogeneity as factors such as stochasticity in gene expression[25,26] and uneven plasmid distribution[12] create cell-to-cell variability in biosensor expression levels that cannot be averaged out across the population and instead, inadvertently introduce biases.

Here, we propose a modular and standardised framework based on robust characterisation of single cells' metabolic states to facilitate the study of isogenic heterogeneity (Fig. 1). We isolate biosensors from the literature which target a range of different core intracellular metabolites and integrate them as functional parts within the widely used *S. cerevisiae* MoClo YTK toolkit[27], thereby establishing an extension to the toolkit which we refer to as YTK-ScBiosense. We then leverage the extended toolkit to systematically quantify single-cell metabolic responses to environmental challenges, specifically investigating acute and gradual depletion of glucose as a primary trigger of differentiation into metabolic subpopulations and examining the impact these subpopulations have on bioproduction heterogeneity. Lycopene and violacein are two naturally occurring pigments with important applications as colouring agents within the textile and food industry, but also as therapeutic agents within healthcare[28–30]. Using their bioproduction as case studies and intracellular pH ($pH_i$) sensing signal as a subpopulation marker, we show depletion of glucose within batch fermentation to induce the emergence of subpopulations with differential production efficiencies. We demonstrate trends in efficiencies to be product-dependent, with cells from the lower $pH_i$ subpopulation displaying improved product accumulation within violacein-producing strains but reduced yields within lycopene-producing strains, and highlight cultivation strategies as a potential method to mitigate subpopulations' impact on bioproduction yields. Collectively, these results demonstrate our framework's versatility within the study of isogenic heterogeneity, providing valuable insights into its environmental triggers, showcasing the role of $pH_i$ and by extension the stress response within the differentiation process, and highlighting its use as a proxy for production efficiency.

## Results and discussion
### Sensing and reporter part selection
Nine different biosensors, each with previously demonstrated functionality in yeast, were selected for the proposed YTK-ScBiosense

toolkit (Supplementary Table 1). They were the following: pHluorin2[31], 2.6 C45-HRRz (referred to as FBPs)[32], pRPL28, pRPL31B, pHXT5[33], pGPD2[34], pYRE[35], yAT1.03[36] and PercevalHR[37] sensing $pH_i$, fructose-1,6-bisphosphate (FBP), growth, NADH, NADPH, absolute ATP, and the ratio of ATP:ADP respectively, to collectively provide a comprehensive overview of a cell's metabolic state. More specifically, $pH_i$ serves as a valuable indicator of cellular stress, with $pH_i$ homeostasis being vital to maintaining cellular function and transient drops in $pH_i$ often playing a crucial role in initiating cells' stress responses[38–40]. Growth is similarly representative of cellular fitness, with its regulation being fundamental to maintaining viability[33]. FBP is an intermediate of the glycolytic pathway whose abundance directly correlates with glycolytic flux, a core component of carbon metabolism[41], while absolute ATP levels and the ratio of ATP:ADP are reflective of a cell's energy status[36,37,42,43]. Lastly, $NAD^+$:NADH and $NADP^+$:NADPH ratios determine the redox balance of cells, which is critical in allowing many enzymatic reactions to occur and thus, in regulating many biological processes[44].

Genetically encoded biosensors can be broadly broken down into two distinct components: a sensing unit which detects and transduces a given input signal, such as the presence of a target metabolite, and a reporter unit, which translates the sensing unit's signal into a measurable output. To integrate our selected sensors as parts within the YTK toolkit[27], we first isolated their core sensing units from literature-obtained constructs and integrated them within Level 0 (L0) YTK backbones, in accordance with each of the sensors' specifications (Supplementary Notes 1). We then screened a selection of 15 commonly used fluorescent proteins (FPs) to identify the ones best suited for use as complementary reporter units (Supplementary Table 2). Specifically, we grouped our selected FPs according to their spectral properties and evaluated their respective in vivo brightness, with bright FPs promoting response strength and, by proxy, functionality of sensing constructs. We observed mTagBFP2, mNeonGreen (referred to as mNeon), eCitrine and mScarletI to consistently display the strongest BFP, GFP, YFP, and RFP signals respectively (Supplementary Fig. 1), and thus integrated them within their own L0 backbones to be used as reporters moving forward. While FP brightness is critical, it is worth noting that other FP properties may also need to be considered for a given application. For example, an FP's pKa should be taken into account when implemented within potentially acidic conditions or its photostability within time-lapse experiments[45].

With each of the sensing units and selected reporters integrated as standardised L0 parts, modular reconstitution of operational sensing constructs was achieved through the hierarchical assembly of promoter, coding sequences (CDS) and terminator parts into transcription units. This modular approach allows for much flexibility in the design of sensing constructs, especially when leveraging the numerous readily available YTK-parts. For instance, separation of the sensing and reporter parts enables interchangeability of reporters, which is paramount to ensure no signal overlap when multiplexing of multiple sensing constructs. Similarly, the large catalogue of available YTK promoter parts facilitates fine-tuning of sensing construct's expression levels, which is especially pertinent when optimisation of its sensitivity (i.e., detection range) and dynamic range (i.e., response strength) is required within a given experimental set-up or strain. Of note, the degree of modularity will be dependent on the specifics of a sensing unit, with, for example, modularity in reporters of FRET and FRET-like sensors being restricted due to their sensing mechanism being intrinsically linked to their fluorescent properties.

### Sensing units part characterisation
To ensure that the isolated sensing parts retained functionality within our YTK-ScBionsense toolkit, each was assembled into operational sensing constructs (Supplementary Fig. 2) and validated at the population-level using assays tailored to the targeted metabolites. mNeon was leveraged as a reporter for TF-based and RNA-based

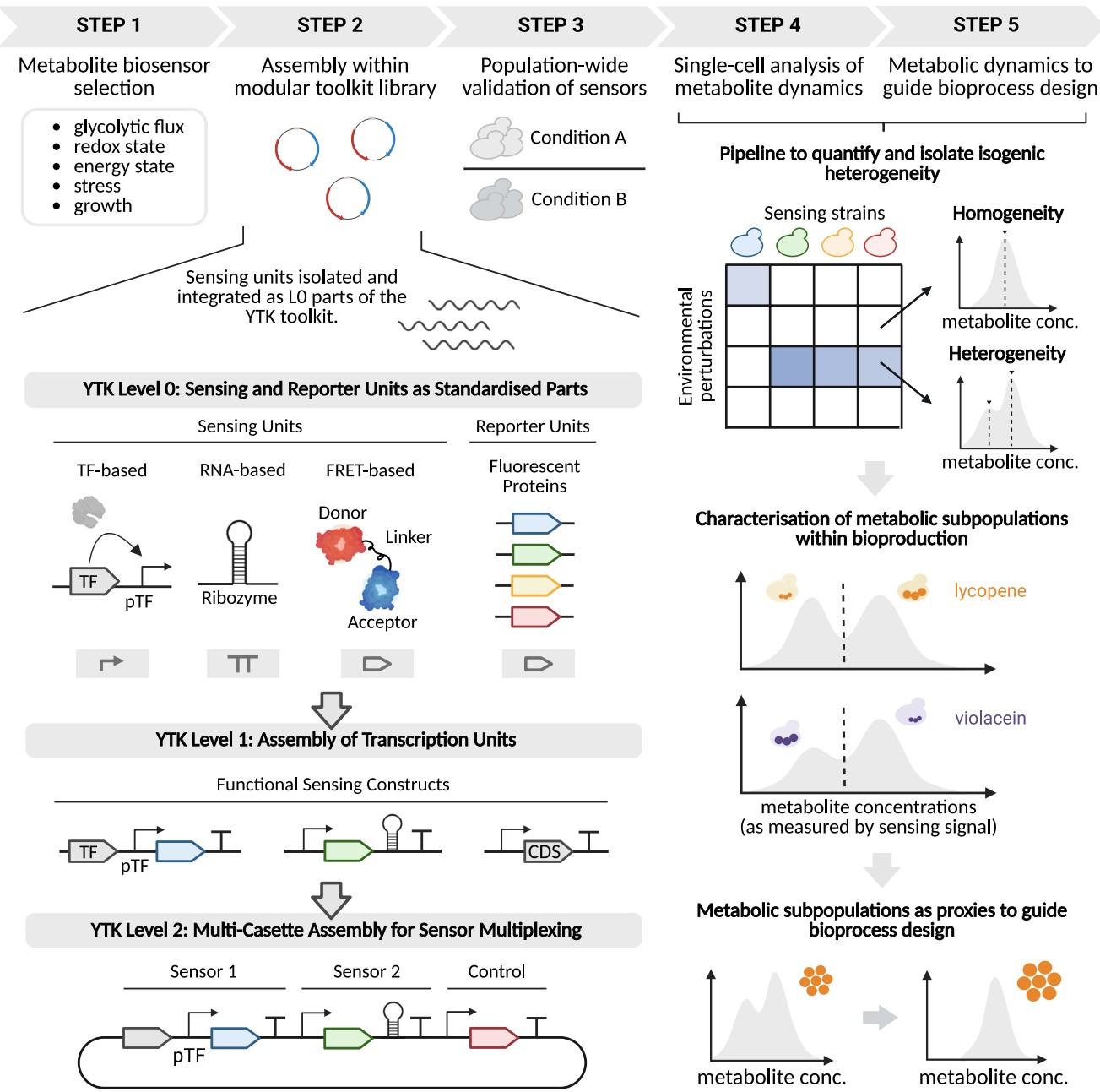

**Fig. 1 | Framework for analysing isogenic heterogeneity using metabolic biosensors.** Sensing units targeting a range of different core intracellular metabolites were isolated from the literature and introduced as Level 0 parts within the YTK toolkit. Combinatorial assembly using pre-characterised YTK parts was then performed to generate operational sensing constructs. Retention of the constructs' sensing abilities was validated at the population-level, while systematic single-cell quantification of sensing signals following exposure to a range of experimental perturbations highlighted the value of this toolkit for quantification and isolation of isogenic heterogeneity. Metabolic subpopulations were then characterised within lycopene- and violacein- producing strains, and used as proxies for high- producing phenotypes to guide bioprocess design. This figure was created in BioRender. Ledesma-Amaro, R. (2025) https://BioRender.com/g89vroh.

constructs, while pTEF1, a robust high-expression promoter, was used for construct expression when applicable. Readout was normalised to mScarletI signal, placed downstream of the sensing unit and under constitutive pTEF2 expression, to correct for any cell-to-cell variation in expression levels which may otherwise bias sensing output. Such precautions were not included for growth sensing constructs as constitutive promoters, including pTEF2, tend to themselves be impacted by growth. This was also not required for FRET and FRET-like constructs as they are ratiometric by design, with variations in their expression levels inherently accounted for in their output. All sensing constructs were integrated within the genome to mitigate any heterogeneity which may arise from uneven plasmid expression[12].

pHluorin2 and pYRE-mNeon, targeting $pH_i$ and NADPH levels respectively, were validated by exposing construct-expressing cells to either a weak acid or diamide treatment. Exposure of cells to weak acids, including sorbic and acetic acid, induces acidification of cells' $pH_i$[46,47], while exposure to diamide, a thiol oxidising agent, depletes their NADPH pools[35]. In both cases, a dose-dependent sensing response was observed, with pHluorin2 signal dropping 38% and 47% upon addition of 5 mM and 10 mM sorbic acid respectively and a strong positive correlation (Pearson coefficient, $\rho = 0.97$) being observed between diamide concentrations and pYRE-mNeon signal 2 h post-treatment, coherent with pYRE expression being induced by reduced NADPH levels (Fig. 2A, B).

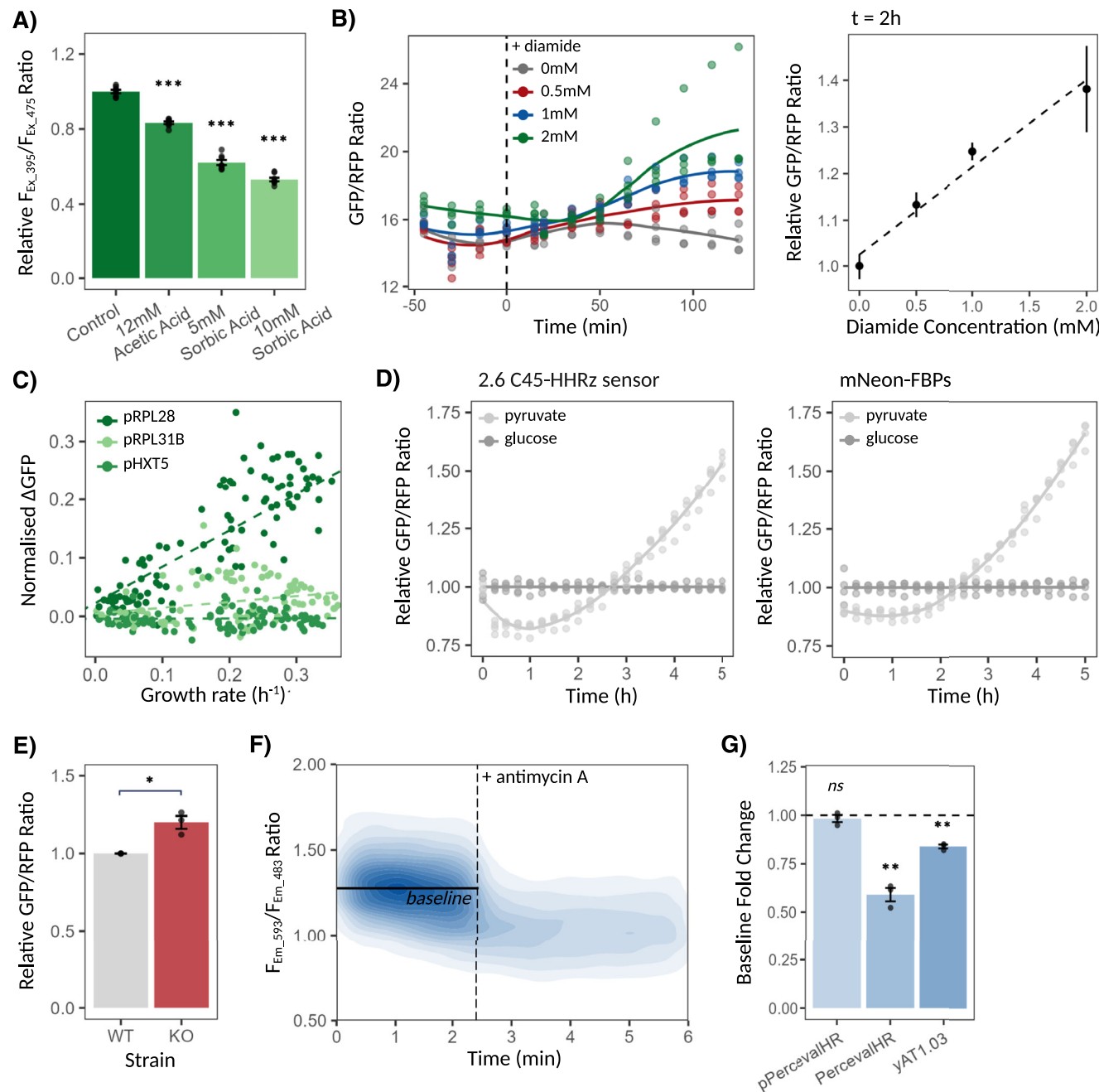

**Fig. 2 | Validation of mNeon sensing constructs functionality. A** Dose-proportional decrease in pHluorin2 signal in response to increasing weak acid exposure ($n = 8$ technical replicates, $p < 0.001$ based on Welch Two Sample $t$-test with control condition). **B** Change in pYRE-mNeon signal following exposure to varying concentrations of diamide with respect to time (left), with resulting correlation between pYRE-mNeon signal, relative to control 0 mM condition, and diamide concentrations 2 h post exposure displayed (right). Dashed line represents fitted linear regression ($n = 4$ technical replicates, $\rho = 0.97$). **C** Scatterplot between growth rate and change in sensing signal, normalised to non-sensing control strain, of pRPL28-, pRPL31B- and pHXT5-mNeon expressing strains at each timepoint of growth curve. Dashed lines display fitted linear regressions. **D** Change in mNeon-FBPs and pFBP-2_6 positive control signals following shift from glucose to pyruvate media as opposed to when shifted back to glucose media. **E** Difference in pGPD2-mNeon signal between WT and *gpd1Δgpd2Δ* KO sensing strains ($n = 3$ biological replicates, $p = 0.041$ based on Welch Two Sample $t$-test). **F** Change in yAT1.03 signal of strains pre-grown on ethanol following exposure to antimycin A inhibitor (AA). Baseline calculated by averaging sensing signal prior to AA exposure. **G** Drop in ATP sensing signals relative to baseline (as calculated in **F**) following exposure to AA. Dotted line represents baseline ($n = 3$ biological replicates, $p < 0.01$ (**) and $p > 0.05$ (*ns*) based on Welch One Sample $t$-test to baseline mean value of 1). Bar plots represent mean ± SEM across specified $n$ replicates. Source data are provided as a Source Data file.

mNeon-FBPs was validated by shifting cells pre-grown on glucose to media with either glucose or pyruvate as a carbon source. Glucose is broken down to pyruvate through the glycolytic pathway as part of both fermentation and respiration. Glucose-grown cells thus have a greater dependency on glycolysis for energy, resulting in higher glycolytic flux, elevated FBP levels and by consequence, reduced FBPs

signal[32,48]. Despite an initial drop, mNeon-FBPs signal in pyruvate-shifted cells steadily increased over time, reaching levels 65% higher than those of glucose-shifted cells 5 h post shift, in accordance with the sensing construct's expected behaviour (Fig. 2D, left). Pyruvate-shifted cells encoding for pFBP-2_6. sensor, Ortega et al. original construct used here as our positive control[32], displayed a similar trend though its

signal increase was more muted, at 53% 5 h post shift, implying a weaker dynamic range (Fig. 2D, right). Control constructs in which the functional FBP-sensitive ribozyme was absent showed no significant differences in sensing signal upon shift, confirming the trends observed to indeed occur as a result of ribozyme activity (Supplementary Fig. 3).

Validation of the ATP-sensing constructs was performed by treating pRS425-PercevalHR (pPercervalHR), PercevalHR, and yAT1.03-expressing cells pre-grown on ethanol with antimycin A (AA), an inhibitor of respiration. pPercevalHR refers to the original plasmid-expressed version of the biosensor, while PercevalHR refers to our genome-integrated variant. PercevalHR and yAT1.03 respectively displayed a 1.69- and 1.19-fold reduction in sensing signal from their pre-exposure baselines upon AA treatment, consistent with ethanol-grown cells primarily relying on respiration for energy generation and thus losing their ability to maintain ATP levels upon treatment (Fig. 2F, G). pPercevalHR, by contrast, showed no significant change in signal following exposure to AA, likely due to the strong noise present within its sensing output (Fig. 2G, Supplementary Fig. 4A). No significant signal change was either observed upon exposure to solvent with no AA for any of the sensing constructs, confirming signal reduction to indeed be due to AA activity (Supplementary Fig. 4B).

Lastly, validation of growth-responsive constructs was performed by growing pRPL28-, pRPL31B-, or pHXT5-mNeon expressing cells to saturation, with growth rate and signal change being recorded at every timepoint of growth curve, while pGPD2-mNeon validation involved cloning of the construct into WT and $gpd1\Delta gpd2\Delta$ KO strains and comparing their resulting sensing signals. While pRPL28-mNeon showed a positive correlation between its signal change and growth rate, in line with expectations, pHXT5-mNeon and pRPL31B-mNeon both yielded signals too weak for a meaningful correlation to be detected and were therefore dropped from further consideration ($\rho$ = 0.50, 0.00, and 0.09 respectively, Fig. 2C). $gpd1\Delta gpd2\Delta$ KO cells encoding for pGPD2-mNeon exhibited a signal 1.20x stronger than that of WT cells, indicating increased NADH levels (Fig. 2E). This aligns with previous studies which have shown $gpd1\Delta gpd2\Delta$ double mutants to exhibit a reduced capacity to convert NADH back to NAD + , leading to accumulation of NADH[49], and thereby confirms pGPD2-mNeon functionality.

Our YTK-ScBiosense toolkit thus retains 7 sensing parts which can be combined with 4 FP reporters, 23 promoters and 13 integration vectors, 3 readily available within the YTK toolkit and another 10 targeting neutral sites from the MYT toolkit[50] (Supplementary Fig. 5). It is worth noting that the evaluation of sensing units in combination with alternate FPs as reporters, that is, mTagBFP2, eCitrine and mScarletI, revealed variability in their performance, thereby highlighting the importance of assaying functionality before use of any newly constructed biosensor (Supplementary Notes 2, Supplementary Figs. 6, 7, 8). The expression of all tested constructs had a minimal impact on cell growth kinetics, suggesting minimal burden (Supplementary Fig. 9).

### Systematic analysis of metabolic environmental responses

As isogenic heterogeneity is primarily associated with promoting survival when faced with environmental perturbations, we next sought to use our sensing constructs to systematically quantify the metabolic response of single cells when shifted from rich YPD media to 10 different media conditions, each presenting a range of different metabolic challenges[51]. These included minimal media with both excess (SMD) and limited (NLIM_PRO, NLIM_GLN, NLIM_UREA, NLIM_NH4) nitrogen, with the nitrogen sources for the latter varying in their preferential utilisation by *S. cerevisiae*[52]. Conditions in which fermentation (YPD, SMD) vs respiration (YPEtOH, SMEtOH) are favoured were also tested, in addition to one carbon starvation condition (CSTRAVE), and one condition in which the TOR signalling pathway, a key regulator of growth[53], was inhibited by rapamycin (RAPA).

Population-level analysis of cells' metabolic response showed a reduction in $pH_i$ upon shift to CSTRAVE and maximal growth following shift to YPD when compared to all other conditions, consistent with previous well-established findings (Fig. 3A, Supplementary Figs. 10, 11)[54–57]. Results from pGPD2 and pYRE also aligned with recent findings on cofactor abundance under respiratory vs fermentative growth[58], with cells displaying altered NADH dynamics upon shift to YPEtOH when compared to YPD, as evidenced by a decrease in pGPD2 signal, while $NADP^+$:NADPH levels remained constant across conditions. Interestingly, we also observed elevated FBP levels, as indicated by reduced mNeon-FBPs signals, upon shift to CSTRAVE and SMEtOH, which may indicate increased gluconeogenesis to compensate for the low glucose levels present within these conditions or, alternatively, could arise as a result of $pH_i$ biasing reporter output. Indeed, the $pH_i$ of stressed cells has been reported to reach pHs as low as 4.7[55] while the pKa of mNeon hovers at 5.4[45]. It is thus possible that the drop in $pH_i$ observed within certain CSTRAVE and SMEtOH-shifted cells may be inducing deprotonation of mNeon, resulting in artificially low and heterogeneous reporter outputs – as was observed within strains encoding for mNeon under unregulated, constitutive expression (Supplementary Fig. 12). mScarletI, by contrast, has a pKa of 4.0[45] and displayed robustness to changes in $pH_i$, prompting us to repeat the shift to these conditions using sensing constructs which made use of mScarletI as a reporter – when feasible – to ensure no bias carried over to our analysis of heterogeneity.

Isogenic cell-to-cell variations can be broadly categorised into micro- vs macro-heterogeneity. Micro-heterogeneity refers to the broad distribution of phenotypes that may occur along a given metric, while macro-heterogeneity refers to the clustering of phenotypes into distinct subpopulations. While micro-heterogeneity is typically quantified using robust coefficient of variance (rCV), a metric of a given distribution's spread[59], macro-heterogeneity may be quantified using the Hartigan's dip test, a measure of multimodality[60] (see Methods). Using the dip test statistic ($D_n$) as a measure of macro-heterogeneity, we observe a uniform metabolic response following shift to most media conditions, as evidenced by shifted cells displaying a unimodal signal distribution with low $D_n$ values across all sensing constructs (Fig. 3B, Supplementary Fig. 10). Deviations did, however, arise, the most significant of which occurred in response to acute depletion of glucose. Indeed, cells shifted to CSTRAVE and SMEtOH, two conditions characterised by sudden glucose starvation-induced stress, consistently differentiated into stable, metabolically distinct subpopulations across multiple metabolites, including $pH_i$ and absolute ATP levels. Cells shifted to YPEtOH did not yield the same differentiation pattern despite the lack of glucose, possibly due to elements of the nutrient-rich YP- media compensating or, alternatively, delaying the onset of heterogeneity. The latter is supported by the observation of a skewed $pH_i$ distribution 8 h post shift to YPEtOH, typically indicative of an emerging subpopulation (Supplementary Fig. 10).

### Characterisation of CSTRAVE-induced $pH_i$ subpopulations

Macro-heterogeneity in response to acute depletion of glucose was most pronounced within pHluorin2's sensing signal and upon shift to CSTRAVE media, prompting us to further investigate glucose-induced subpopulation differentiation using these conditions as a case study. Clustering of cells based on their $pH_i$, as determined by pHluorin2 signal, revealed two dominant subpopulations, with approximately ~20% of cells exhibiting a significant drop in $pH_i$ post shift (p1), while the remaining majority maintained $pH_i$ levels closer to those observed under optimal conditions (p2). Co-expression of pHluorin2 and pRPL28-mScarletI sensing constructs showed the subpopulations to also follow distinct growth patterns, with p2 displaying greater growth capacity compared to p1 4 h post shift (pRPL28 signal = 0.94 and 0.85 a.u. respectively, Fig. 3C). p2's pRPL28-mScarletI signal later decreased to match that of p1 at 8 h post shift, suggesting p2's ability to maintain

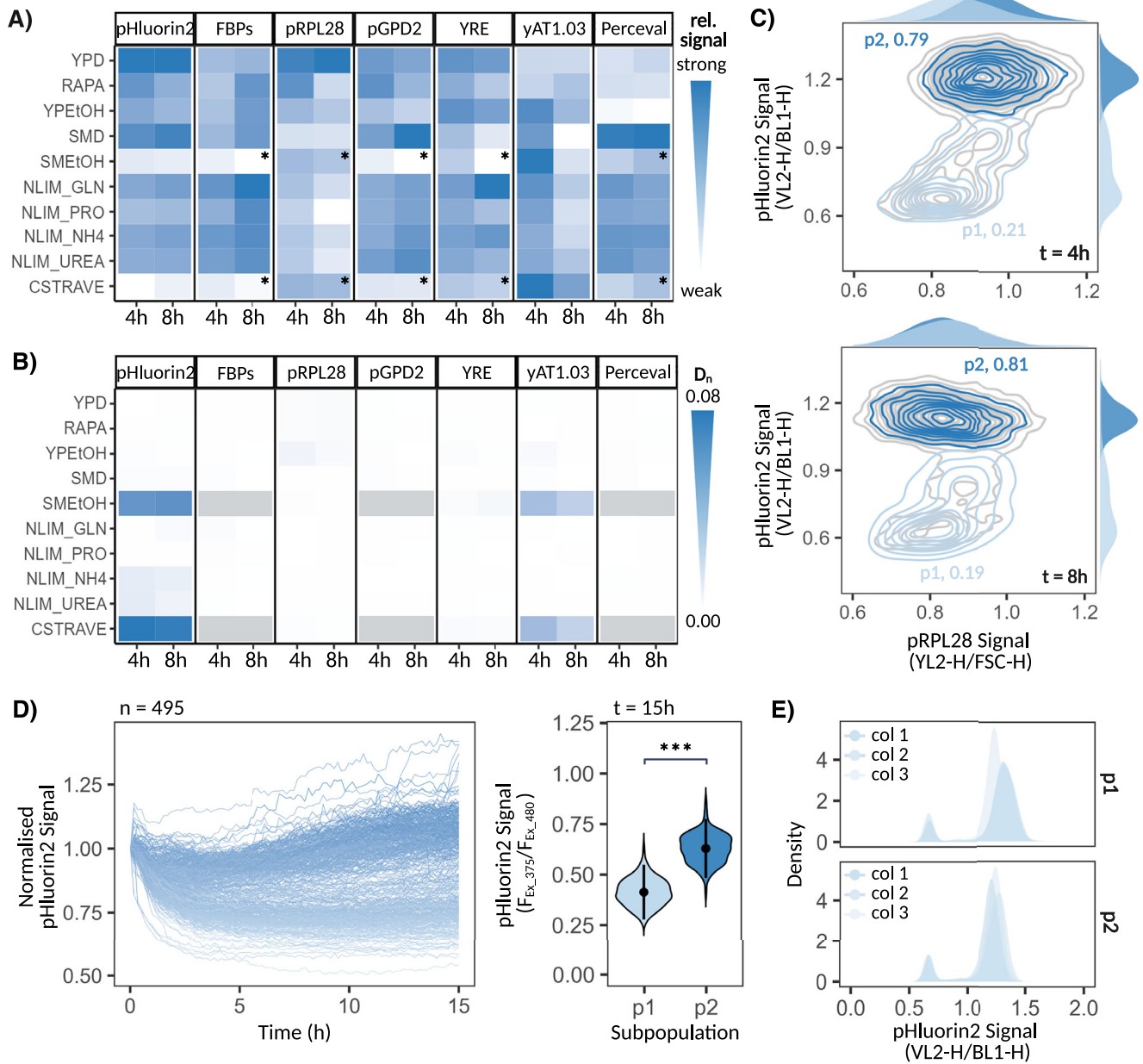

**Fig. 3 | Metabolic response to environmental perturbations. A** Relative population-averaged sensing signal and (**B**) signal macro-heterogeneity, as quantified by the Hartigan's dip test statistic, 4 h vs 8 h post shift from rich YPD media to various media conditions, averaged across 3 biological replicates. * denotes sensing values which may be influenced by pH_i, while grey squares denote constructs for which the use of mScarletI as an alternative, pH-robust sensing reporter was not available. **C** Counterplot showing the relationship between pHluorin2 and pRPL28-mScarletI signal 4 h (top) and 8 h (bottom) post shift to CSTRAVE media, with the relative frequency of each subpopulation displayed. **D** Tracked changes in

pHluorin2 signal of single cells post shift to CSTRAVE media (left), with distribution in p1 and p2 sensing signals at assay endpoint displayed (right, $p < 2.2e\text{-}16$ based on Welch Two Sample $t$-test). Tracked signals normalised to initial pH_i (at t = 0 h) to reduce noise and allow for inter-cell comparison of pH_i dynamics. **E** pHluorin2-expressing cells were sorted into p1 vs p2 subpopulations 4 h post shift to CSTRAVE media and plated on fresh YPD plates to reset phenotype. 3 separate colony isolates from each plated subpopulation were then shifted back to CSTRAVE media, with the resulting pHluorin2 signal distribution displayed. Source data are provided as a Source Data file.

growth could not be sustained within prolonged starvation conditions. A closer look at differentiation dynamics using time-lapse microscopy revealed all cells to experience a slight drop in pH_i immediately post shift, with the pH_i of p2 cells bouncing back as time progressed while that of p1 cells continued to decrease, until it stabilised at around 4 h post shift (Fig. 3D, Supplementary Fig. 13A). These results also showed the subpopulations to remain stable throughout the observation period, persisting 15 h post shift with no cells transitioning between the subpopulations. Sorting of cells based on their pHluorin2 signal post shift further revealed p2 to have increased viability, as evidenced by a

higher CFU count when plated on rich YPD media (424 and 132 CFUs for p2 and p1 respectively), and greater persistence, with p2 cells showing a 3.35-fold increase in survival rate when stored in PBS for 7 days (26.5% vs 7.63% survival rate for p2 vs p1 respectively, Supplementary Fig. 13B, C). p2 cells also displayed better regrowth dynamics when shifted back to rich YPD media, as demonstrated by them reaching saturation faster, thereby suggesting an improved capacity to resume growth when the right conditions reappear (Supplementary Fig. 13D). Lastly, the sorted subpopulations were plated on rich YPD media to reset their phenotype before being shifted back to CSTRAVE

media. Doing so resulted in the emergence of p1 and p2 within all of the colony isolates tested, irrespective of the subpopulation's history, thereby confirming the subpopulations to occur as a result of isogenic heterogeneity and not recurring spontaneous mutations (Fig. 3E).

Together, these results align with previous reports of glucose starvation-induced subpopulation differentiation. For instance, in their model of cellular differentiation, Davidson et al. posit that cells enter either a state of quiescence or non-quiescence upon depletion of glucose, with quiescent cells retaining greater reproductive capabilities, in line with the behaviour identified from p2[61]. Bagamery et al.'s bet-hedging hypothesis similarly describes two distinct phenotypic responses to glucose starvation, correlating fermentation pre-glucose starvation to a drop in pH$_i$, collapsed mitochondria and cellular arrest[6,62]. This is in contrast to cells undergoing respiration which, despite having a slower growth rate under optimal conditions, are posited to have a greater ability to adapt and thus resume growth upon depletion of glucose. These observations parallel the behaviour observed from p1 and p2 subpopulations, with p2 not only exhibiting improved growth but also displaying greater viability and persistence in response to glucose depletion.

### pH$_i$ subpopulation dynamics within batch cultivation

Fermentation of industrially relevant products often requires extended cultivation, beyond glucose depletion and density saturation, to allow for product accumulation and yield maximisation[63–65]. Accordingly, and given the widespread occurrence of low-producing subpopulations within bioproduction, we next pondered whether the pH$_i$ subpopulations we previously observed would also emerge within batch cultivation as a result of gradual glucose depletion and what impact they may have on the production efficiency of microbial cell factories (MCFs). To that end, pHluorin2 was introduced within strains encoding for either the violacein- (Fig. 4A) or lycopene- (Fig. 4B) production pathway and cultivated in flasks mimicking batch fermentation conditions. Fully differentiated subpopulations appeared within both MCFs following 24 h of flask batch fermentation and persisted throughout the cultivation process (Fig. 4D, Supplementary Fig. 14). Interestingly, the onset of differentiation occurred earlier within violacein MCFs, with subpopulations emerging following 16 h of cultivation, in direct correlation with cultures reaching maximum carrying capacity and glucose levels being fully consumed (Supplementary Fig. 15). Lycopene MCFs, by contrast, grew at a slower rate, with remaining glucose levels being 41 times that of violacein-producing cultures at 16 h (0.83 g L$^{-1}$ as opposed to 0.02 g L$^{-1}$), seemingly resulting in delayed differentiation by comparison. No subpopulations emerged from fed-batch-like flask fermentation, characterised by glucose pulses being periodically supplemented to cultures, thereby reinforcing the importance of glucose depletion within the differentiation process (Supplementary Fig. 14).

Lycopene- and violacein-producing strains both exhibited subpopulation dynamics comparable to those observed following shift to CSTRAVE media, with differentiation triggering a sharp decline in p1's pH$_i$ whereas p2 maintained levels closer to those of the population pre-differentiation (Supplementary Fig. 16). p1's signal remained steady upon differentiation, regardless of the product or sampling time, whereas p2's signal showed a gradual decline, dropping from 1.23 and 1.29 to 1.00 and 1.06 a.u. following 24 h to 72 h of cultivation within violacein and lycopene MCFs respectively (Supplementary Fig. 14). Curiously, while the relative frequency of p1 when compared to p2 remained stable throughout the cultivation process, it differed between the products, persisting at approximately ~20% and ~40% of the total violacein and lycopene producing population, respectively. This discrepancy may arise from a range of factors, most notable of which is the differences in metabolic loads imposed by the two production pathways, with them pulling resources away from distinct metabolic pathways, which could promote the activation of different stress responses or influence the cell's predispositions to enter a stress state.

### pH$_i$ subpopulations as a source of production heterogeneity

Clustering of violacein-producing cells based on their pHluorin2 signal showed p1 to directly correlate with increased violacein production, as indicated by increased detection of violacein autofluorescence. (Fig. 4E–G). Indeed, p1 displayed an enhanced ability to produce and accumulate violacein, with its autofluorescence measurements increasing 7.36-fold from 0.28 a.u. to 2.06 a.u. following 24 h and 72 h of cultivation respectively (Fig. 4F). The accumulation of violacein within p2 was, by contrast, much less pronounced, showing an 82% reduction in efficiency when compared to p1 at the end of the cultivation period. Microscopy analysis of pHluorin2-sorted subpopulations confirmed these findings, with violacein autofluorescence being detected exclusively within cells belonging to the p1 subpopulation, but not from p2 (Fig. 4G). Strikingly, the trend observed within lycopene-producing cells was reversed (Fig. 4I). Single-cell Raman microscopy of sorted subpopulations revealed lycopene peaks to be 3.71x more pronounced within p2 compared to p1, with a peak area of 0.074 a.u. for p2 compared to 0.020 a.u. for p1 (Fig. 4I left). Consistent with these measurements, the pellet of p1 was noticeably lighter in colour relative to that of p2 (Fig. 4I right). The mixed population, containing both the p1 and p2 subpopulations, showed a 40% reduction in lycopene production when compared to p2, in line with the subpopulation ratios and further highlighting the importance of implementing strategies aimed at eliminating p1 for yield optimisation.

Lycopene is derived from acetyl-coA, an intermediate of respiration[66], while violacein is derived from L-tryptophan, an amino acid which has been shown to promote tolerance to a range of different stressors[67–69]. The distinct behaviours in product accumulation may thus arise from the distinct metabolic responses between the subpopulations, with p2 displaying increased mitochondrial staining, indicative of enhanced respiratory capabilities (Supplementary Fig. 17), and p1 displaying greater cellular stress, as indicated by its lower pH$_i$.

### Modulating subpopulation dynamics to promote production

Having demonstrated the presence of subpopulations to impair overall production efficiency of industrially relevant metabolites, we next decided to investigate whether the implementation of informed cultivation conditions could be used to fine-tune subpopulation ratios —with the goal of optimising production levels. Specifically, we explored the potential of galactose as a carbon source to alleviate the burden imposed by p1 on lycopene production, by favouring differentiation to p2. Galactose-grown cells have been shown to display reduced overflow metabolism and increased respiration levels[70,71], which we hypothesised could contribute to increased p2 frequency, in line with previous observations by Bagamery et al.[6]. To test this hypothesis, lycopene-producing strains were grown within deep-well plates with either 2% galactose or 2% glucose as a carbon source, with the resulting subpopulation dynamics and lycopene production analysed. Results showed the relative frequency of p1 to drop significantly under galactose conditions, with p1 comprising 69.8% of the total cell population when grown on glucose and dropping to 31.6% under galactose cultivation conditions (Fig. 5A, E). This altered subpopulation dynamics paralleled the changes observed in single-cell lycopene production under both culture conditions, whereby galactose cultivation prompted a shift from low- to high- producing phenotypes when compared to glucose cultivation (Fig. 5B). The decrease in p1 frequency also coincided with enhanced lycopene production, reaching 2.12 mg L$^{-1}$ within galactose cultivation, a 1.94-fold increase from the 1.09 mg L$^{-1}$ seen under glucose cultivation, and enhanced production relative to biomass, increasing from 0.021 mg g$^{-1}$ to 0.035 mg g$^{-1}$ under glucose and galactose cultivation respectively (Fig. 5C, D). This

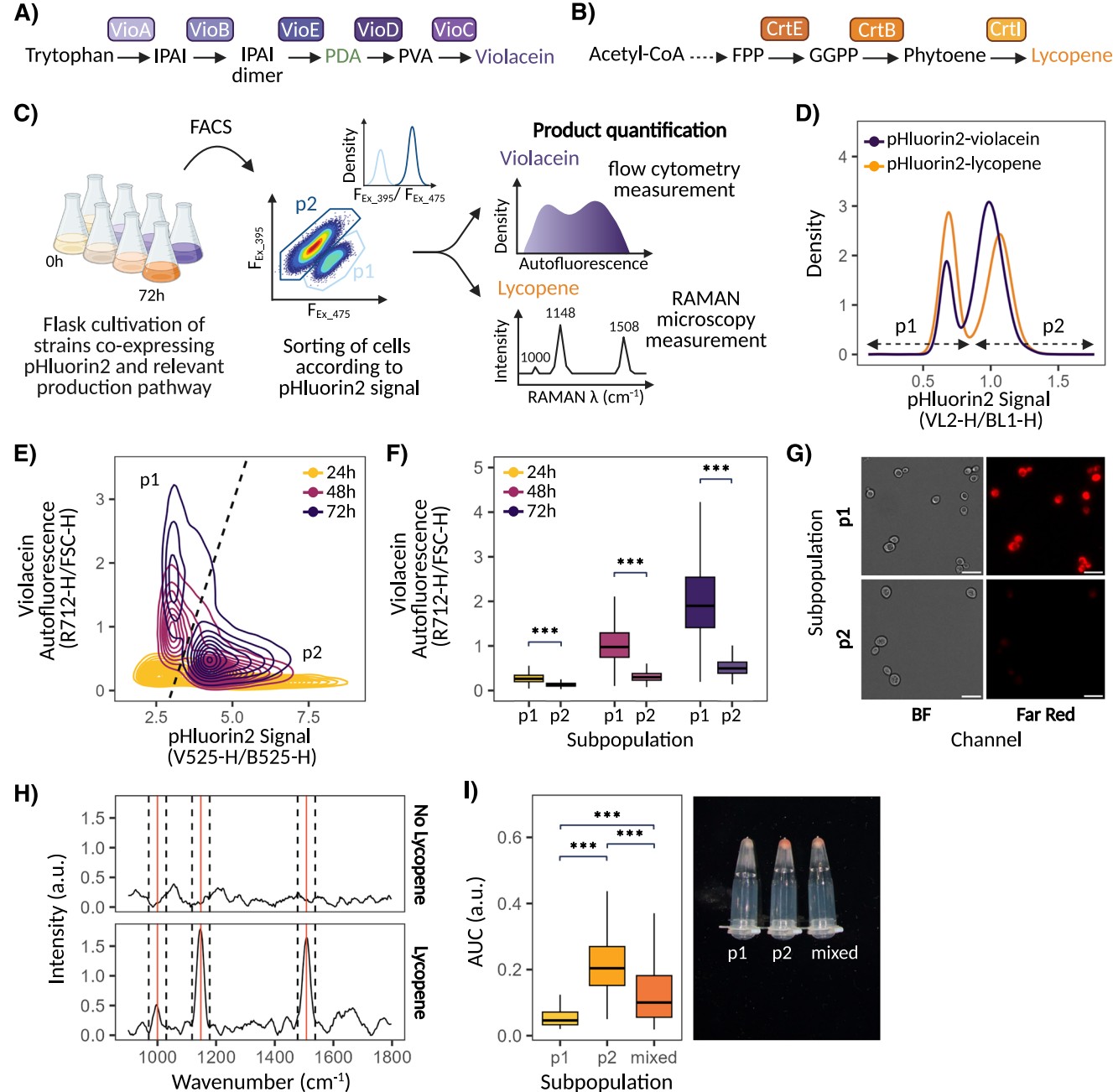

**Fig. 4 | pH subpopulations as a source of production heterogeneity. A** Violacein and (**B**) lycopene production pathways. **C** Pipeline used to isolate and quantify production within pH$_i$ subpopulations of violacein and lycopene MCFs. **D** pHluorin2 signal distribution of MCFs following 72 h of flask cultivation. **E** Counterplot showing the evolution between pHluorin2 signal and violacein autofluorescence of violacein-producing cells over time. Dashed line denotes separation between the p1 and p2 subpopulations. **F** Comparison of violacein production, as determined by autofluorescence, within p1 and p2 subpopulations following 24 h, 48 h, and 72 h of flask cultivation (*n* = 10,000 cells). **G** Multi-channel imaging of p1 and p2 subpopulations of violacein-producing strains following 72 h of flask cultivation. Far-red channel leveraged for detection of violacein auto-fluorescence. Scale bars represent 10 µm. **H** Raman spectrum of non-producing and lycopene-producing cells. Red line indicates lycopene peak location while dotted lines display peak range used to calculate area under the curve (AUC). **I** Comparison of lycopene production, as determined by Raman peaks AUC, of p1 (*n* = 298 cells), p2 (*n* = 620 cells) and mixed (i.e., p1 and p2, *n* = 586 cells) subpopulations of lycopene-producing strains following 72 h of flask cultivation (right). Pellet colour of sorted subpopulations also displayed (left). Box plots centre marks represent median subpopulation values while their hinges mark lower and upper quartiles. Whiskers show values that fall within 1.5x of the interquartile range. Significance scores denote *p*-value, with *p* < 0.001 (***) based on Mann–Whitney *U* test. a.u. denotes arbitrary units. This figure was created in BioRender. Ledesma-Amaro, R. (2025) https://BioRender.com/g89vroh. Source data are provided as a Source Data file.

carbon source-dependent pattern persisted within micro-bioreactor cultivation and under pH buffering conditions, confirming robustness of the observed subpopulation dynamics, including to changes in external pH (Fig. 5E). Trends in production under these conditions were also consistent with those observed within deep-well plates, although the reduced culture volumes and biomass of micro-

bioreactors promoted variability within the resulting production measurements (Supplementary Fig. 18A). Of note, switching carbon source from glucose to galactose had a similar effect of the pH$_i$ dynamics of violacein-producing populations, with their relative p1 frequencies dropping 22% from glucose to galactose cultivation (Supplementary Fig. 18B).

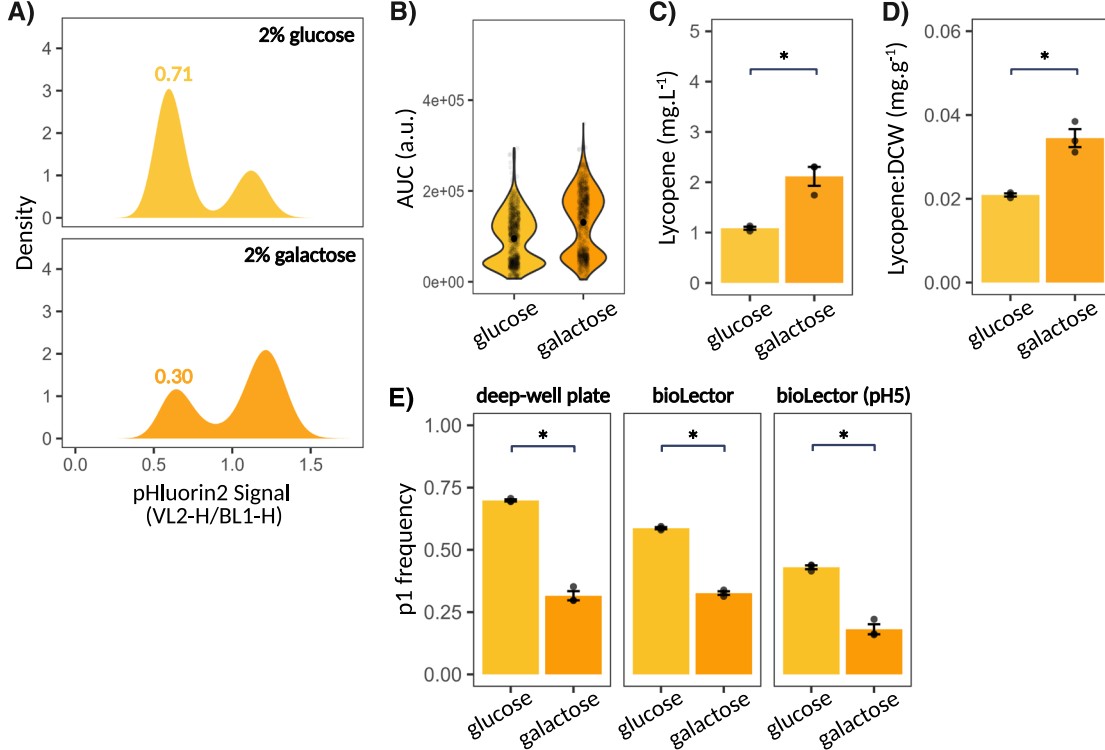

**Fig. 5 | Modulation of subpopulation dynamics through informed cultivation strategies. A** Effect switching carbon source from glucose to galactose had on the $pH_i$ dynamics, as determined by pHluorin2 signal, of lycopene-producing strains following 72 h of cultivation. Relative p1 frequency displayed. Resulting change in (**B**) single-cell lycopene production dynamics, as determined by Raman peak AUC ($n = 807$, $n = 765$ cells for glucose and galactose conditions respectively), (**C**) lycopene production, as measured using HPLC, and (**D**) lycopene production relative to biomass also quantified. Dry cell weight (DCW) used as a measure of biomass. **E** Carbon source-dependent shift in subpopulation dynamics conserved across deep-well plate, BioLector micro-bioreactor, and BioLector with buffering at pH = 5.0 cultivation. Bar plots represent mean ± SEM across 3 biological replicates. Significance scores denote $p$-value, with $p < 0.05$ (*) based on Welch Two Sample $t$-test. a.u. denotes arbitrary units. Source data are provided as a Source Data file.

Together, these results exemplify how tuning subpopulation dynamics towards to a desired phenotype may improve production efficiency, and thereby highlight the importance of understanding these subpopulations and their differentiation mechanism to inform the cultivation and engineering of better producing strains. This may be all the more crucial to take into account during scale-up as uneven mixing within bioreactors creates micro-environments that are deprived of glucose, which may be sufficient to trigger differentiation and thus further amplify the phenomenon[72].

In conclusion, we report a standardised and modular framework which leverages the YTK-ScBiosense toolkit, comprised of 7 different sensing parts, each targeting different core intracellular metabolites, to enable robust characterisation of single cell's metabolic state. We demonstrated this framework's ability to not only systematically quantify single cells' metabolic responses to environmental perturbations but also to identify and isolate metabolically distinct subpopulations, with depletion of glucose in particular emerging as a strong trigger of subpopulation differentiation. Further characterisation of the observed subpopulations using $pH_i$ as a subpopulation marker corroborated previous evidence of a bet-hedging strategy, and revealed $pH_i$ subpopulations as drivers of bioproduction heterogeneity and markers of bioproduction efficiency – thereby bypassing the need for single-cell product quantification to guide bioprocess design. Lastly, we highlight fine-tuning of cultivation strategies as a method to modulate subpopulation dynamics towards a desired phenotype, correlating this change in subpopulation abundance with improved bioproduction yields. Future multi-omics works aimed at elucidating the documented divergent product-dependent responses and their underlying mechanisms – including to account for potential reversible genetic mutations – could not only inform additional cultivation strategies but also engineering targets to enhance both strain productivity and stability.

## Method

### Strains and growth conditions

The *S. cerevisiae* strain used throughout this study was BY4741 (*MATa his3Δ1 leu2Δ0 met15Δ0 ura3Δ0*). CEN.PK2-1C WT (*MATa his3D1 leu2-3_112 ura3-52 trp1-289 MAL2-8c SUC2*) and its *gpd1Δgpd2Δ* KO (*gpd1::TRP1 gpd2::LEU2 his3::HIS3*) were used for characterisation of pGPD2 sensing constructs. With the exception of the shift experiments and unless stated otherwise, strains were grown on either rich YPD media – 20 g L⁻¹ peptone (Merck), 10 g L⁻¹ yeast extract (Sigma), 2% glucose (VWR) – or selective SC-ULH media – 6.7 g L⁻¹ yeast nitrogen base without amino acids (Sigma), 1.4 g L⁻¹ yeast synthetic dropout supplement lacking uracil, leucine and histidine (Sigma), 1% glucose.

### Assembly of sensing constructs

Plasmids were constructed using Golden Gate assembly, as outlined by Lee et al.[27]. Fluorescent proteins and sensing units of interest were isolated from either genomic DNA (pRPL28, pRPL31B, pHXT5 and pGPD2), pCfB4170 (Addgene: #126750, pYRE, mKate2), pDRF1-GW yAT1.03 (Addgene: #132781, yAT1.03v1), pRS425-PercevalHR (NBRP: BYP9646, PercevalHR), pFBP-2_6.sensor (Addgene: #162800), pCggRO-reporter (Addgene: #124582, eCitrine), gblocks (sequences from Mahon et al.[31], pHluorin2, and from Shaner et al.[73], mNeonGreen), or plasmids readily available within the RLA collection (all remaining FPs), via PCR amplification using primers encoding for BsmBI restriction site overhangs. Amplified sequences were purified using a PCR

clean up kit (NEB) and assembled within their corresponding back-bones via L0 Golden Gate reactions (Supplementary Notes 1). Golden Gate reactions were prepared as follows: 50 fmol of each DNA insert, 25 fmol of backbone, 1 μL T4 DNA ligase buffer (NEB), 0.5 μL T7 DNA ligase (NEB), 0.5 μL BsmBI restriction enzyme (NEB), and ddH$_2$O (Invitrogen) to bring the final volume to 10 μL, and incubated using the following thermocycler program: 30 cycles alternating between digestion and ligation (42 °C for 2 min, 16 °C for 5 min) followed by a final digestion step (60 °C for 10 min) and a heat inactivation step (80 °C for 10 min). They were then transformed into *E. coli* using a standard heat shock transformation protocol, with assembled plasmids later extracted via miniprep using QIAprep spin miniprep kit (QIAGEN). Of note, mTurquoise2, Venus, and mRuby2 were already available as L0 parts of the YTK toolkit[27].

Higher order assembly of transcriptions units and multi-cassette constructs were completed using L1 and L2 Golden Gate reactions. These were prepared identically to L0 reactions, with the exception of DNA inserts and backbones which were replaced by L0 parts or L1 cassettes for L1 and L2 inserts respectively, and by integration vectors targeting desired integration loci for the backbone. BsmBI was also replaced by BsaI (NEB) for L1 reactions. When multi-cassette assembly was desired, transcription units were first cloned into pre-assembled intermediary vectors containing the appropriate combination of ConL sequences.

A summary of all L0 parts constructed for the YTK- ScBionsense toolkit can be found in Supplementary Table 2, while final sensing constructs and sensing strains are available within Supplementary Data 1 and 2.

## FP brightness measurements
Single FP-expressing colonies were grown to saturation in SC-ULH, diluted 1:50 in 2 mL of fresh medium, and incubated at 30 °C for 4 h, with 250 rpm shaking, until an OD ~ 0.6 was reached. 30 μL of culture was then diluted in 200 μL PBS (Invitrogen) and analysed using Attune NxT flow cytometer (ThermoFisher), while another 500 μL was pelleted, resuspended in 5 μL PBS, and transferred to microscopy glass slides for analysis on Eclipse Ti microscope fitter with a Hamamatsu ORCA Flash4 Camera with CoolLED p300 light source (Nikon). The Attune's VL1-H, BL1-H, and YL2-H channels and the Nikon's DAPI, GFP, and Cy3 filter cubes were used for BFP, GFP, and RFP data collection respectively. Brightfield microscopy images were also collected for cell segmentation purposes, performed using a pipeline adapted from YeaZ model[74]. For microplate measurements, overnight cultures were diluted to an OD of 0.1 in 150 μL of fresh media within wells of black μClear 96-well plates (Greiner). Plates were then sealed with Breathe-Easy membranes (Sigma) and incubated within Spark microplate reader (Tecan), set at 30 °C with 200 rpm orbital shaking. OD$_{600}$ and fluorescence measurements were taken following 20 h of incubation, with 400 nm (20), 485 nm (20), 560 nm (20) excitation and 465 nm (35), 520 nm (10) and 620 nm (20) emission filter settings being used for BFP, GFP and RFP detection respectively.

## Functionality assays
For microplate perturbation, KO, and growth assays, sensing isolates were grown to saturation in SC-ULH before being diluted to an OD of 0.1 in 150 μL of fresh, pre-aliquoted media within wells of 96-well plate. Plates were then sealed and set up within microplate reader (30 °C, 200 rpm orbital shaking), with OD$_{600}$ and fluorescence at excitation 400 nm (20) and 485 nm (20) and emission 520 nm (10) – corresponding to pHluorin2 signal – on top of regular BFP, GFP and RFP measurements recorded every 30 min. At OD ~ 0.4, equivalent to exponential phase, sorbic acid (final concentration 10 mM and 5 mM) and acetic acid (12 mM) was added to pHluorin2-expressing strains, while diamide (final concentration 0, 0.5, 1 and 2 mM) was added to pYRE-expressing strains. Plates were then placed back into microplate

reader and measurements pursued every 15 min for up to two hours. To obtain growth dynamics and for validation of pGPD2-expressing strains, no interruption or treatment was introduced and, instead, measurements continued until saturation was reached. Exponential phase of pGPD2-expressing strains was determined post-experiment from analysis of growth curves, while stationary phase points were excluded from analysis of pRPL28-, pRPL31B- and pHXT5- results as to avoid skewing data with an abundance of points at μ ≈ 0. Negative growth rates were also excluded from pRPL28-, pRPL31B-, and pHXT5-results as these indicate cell death, which could bias the sensing results.

For condition shift assays, single FBPs-expressing colonies were grown to saturation in SC-ULH, diluted 1:50 in 2 mL of fresh medium and incubated at 30 °C for 4 h, with 250 rpm shaking, until an OD ~ 0.6 was reached. Cells were then pelleted, washed twice in PBS, and resuspended to an OD of 0.1 in fresh SC-ULH media with either 1% glucose or 1% pyruvate (Sigma) as a carbon source. 150 μL of resuspended culture was then aliquoted into wells of 96-well plate, which was sealed and set up within microplate reader (30 °C, 200 rpm orbital shaking). OD$_{600}$, BFP, GFP, and RFP fluorescence measurements were recorded every 15 min for up to 5 h.

Lastly, tracking of ATP levels in response to AA followed a protocol adapted from Luzia et al.[75]. Single ATP-sensing colonies were grown in SC-ULH with 1% EtOH (Fisher Scientific) for 48 h, to let cells acclimate to their new carbon source, before being passaged 1:20 in 2 mL fresh medium for another 24 h. 120 μL of culture was then sampled, diluted in 800 μL fresh media, and analysed using the Attune NxT flow cytometer. VL1-H, VL2-H, VL3-H and BL1-H channels were used for signal detection, with VL3-H:VL1-H and BL1-H:VL2-H corresponding to yAT1.03 and PercevalHR signals respectively. Flow rate was set to 25 μL/min and constrained to 2000 events/sec. Samples were left to run for 2 min to determine baseline signal before being treated with 100 μL of 0.5 mM AA (Sigma) diluted in 5% EtOH. Samples were then left to run for another 5 min to track signal change. In between samples, 1 × 2 min and 2 × 2 min rinses were performed with 100% EtOH and MQ H$_2$O, respectively, to wash off any remaining AA and EtOH.

When applicable, fluorescent measurements were normalised to OD$_{600}$, and fold increase to control condition was calculated. This allowed for response magnitude of different sensing construct variants to be compared.

## Medium shift
Single sensing colonies were grown to saturation in YPD before being diluted to an OD of 0.1 in 5 mL of fresh media and left to incubate at 30 °C for 4 h, with 250 rpm shaking. Cells were then pelleted, washed twice in PBS, and resuspended to an OD of 20 in PBS. 10 μL of resuspended culture was then diluted in 2 mL of condition media (YPD, RAPA, YPEtOH, SMD, SMEtOH, NLIM_PRO, NLIM_GLN, NLIM_NH4, NLIM_UREA, CSTRAVE) within 14 mL glass culture tubes, and placed back to incubate at 30 °C. SM- media were composed of 6.7 g L$^{-1}$ YNB base without amino acid and 38 mM (NH$_4$)$_2$SO$_4$ (Sigma). NLIM- media were composed of 1.7 g L$^{-1}$ YNB base without amino acid or ammonium sulphate (Sigma), supplemented with 0.8 mM of the relevant nitrogen source, while CSTRAVE media was composed of the same base with 1 mM (NH$_4$)$_2$SO$_4$ as a nitrogen source. 1% glucose or 1% EtOH was used as a carbon source, with the exception of CSTRAVE, for which no carbon source was provided. 4 h and 8 h post shift, 20 μL of cultures were aliquoted and resuspended in 200 μL of PBS. Fluorescence of 10,000 events was quantified using Attune NxT, using a flow rate of 100 μL/min. pHluorin2 signal was determined using VL2-H:BL1-H ratio, while BL1-H:YL2-H enabled detection of mNeon-FBPs, pGPD2-mNeon, and pYRE-mNeon signals. pRPL28 signal (BL1-H or YL2-H when multiplexed with pHluorin2) was normalised to cell size, as determined by FSC-H, while yAT1.03 and PercevalHR signals were quantified using VL3-H:VL1-H and BL1-H:VL2-H respectively. YRE-mScarletI signal (YL2-

H) in CSTRAVE and SMEtOH-shifted cells was also normalised to cell size to mitigate bias from $pH_i$ effects on mNeon control output, with cell size displaying a direct correlation with constitutive FP expression (Supplementary Fig. 12). For RAPA condition, cells were first grown in YPD, with 200 ng.mL$^{-1}$ rapamycin (Sigma) treatment added 30 min prior to 4 h timepoint.

## Timelapse microscopy

CSTRAVE-shifted pHluorin2-expressing cells were immediately transferred to black μClear 96-well plates and coated with concanavalin A to restrict cell movement. Cells were then imaged every 10 min for a total of 15 h using the Eclipse Ti2 Twin-Cam-TIRF microscope fitted with two Prime 95B cameras with CooLED 4000 light source (Nikon) and an environmental chamber to maintain temperature at 30 °C. Resulting pHluorin2 signal was quantified by overlaying emission signals detected at 535 nm following excitation at 375 nm and 480 nm. Signal from single cells were tracked using the TrackMate plugin in ImageJ/Fiji, using the LoG (Laplacian of Gaussian) detector for cell segmentation and a simple LAP tracker.

## Subpopulation characterisation

Sorting of pHluorin2-expressing cells into p1 and p2 subpopulations was performed using the CytoFLEX SRT Cell Sorter (Beckman Coulter), based on manually defined FSC-A:FSC-H gates (filtering of debris, doublet, and budding yeast) and V525-H:B525-H fluorescence gates (pHluorin2 subpopulation classification). Analysis of 5,000 events was completed prior to sorting to allow for gate adjustment when necessary. To obtain CFU counts, 1,000 cells per subpopulation were sorted, resuspended in 100 μL of PBS, plated on YPD, and incubated at 30 °C overnight. Resulting colony count performed using Reshape Biotech. For chronological lifespan assay, 500,000 cells per subpopulation were sorted into distinct 5 mL tubes, sealed with parafilm to prevent evaporation, and stored at 30 °C. Viability of cells after 0, 3, and 7 days was determined by quantifying permeability to propidium iodide (PI, Merck). 1 μL of 1 mg/mL PI was added to 200 μL of sorted cell sample, with the resulting fluorescence being analysed using Attune NxT (YL2-H). Live (cells at exponential phase) and dead (cells resuspended in 70% EtOH and incubated at room temperature for at least 30 min) controls were used to determine fluorescence thresholds of viability. To determine regrowth dynamics, 10,000 cells per subpopulation were directly sorted into 8 wells of 96-well plate containing 200 μL of pre-aliquoted YPD. Empty well buffers were placed between each sorted subpopulation, ensuring no cross-contamination due to splashing. Plates were then sealed using a Breathe-Easy membrane and analysed using the CLARIOstar microplate reader (BMG Labtech), with OD$_{600}$ being recorded every 30 min for 24 h. Incubation was set at 30 °C, with 250 rpm double orbital shaking. Lastly, to analyse re-shift dynamics, 1,000 cells were sorted per subpopulation, resuspended in 100 μL of PBS, before being plated on YPD. Single colony was then inoculated in YPD to be shifted back to CSTRAVE media, as described in "Medium shift" section.

## Production strain cultivation

Lycopene and violacein production pathways were introduced within pHluorin2-expressing strains, with resulting strains detailed in Supplementary Data 3. Flask fermentation was achieved by inoculating single colony isolates to saturation before diluting them to an OD of 0.1 in 20 mL of fresh YPD within 100 mL flasks. Flasks were then placed to incubate at 30 °C, with 250 rpm shaking. Every 24 h, 10 g L$^{-1}$ glucose was added to fed-batched -like culture flasks while batched culture flasks remained untouched. For determination of heterogeneity onset, 160 μL of culture was sampled at each of the specified timepoint, with 10 μL being diluted in 200 μL PBS and analysed for pHluorin2 signal quantification using Attune NxT (VL2-H:BL1-H), 50 μL diluted in 950 μL fresh YPD for OD$_{600}$ recording using spectrophotometer, and the

supernatant of the remaining 100 μL extracted and analysed for glucose quantification using HPLC.

To compare impact of glucose vs galactose cultivation, deep-well plates and micro-bioreactor were leveraged. Single colony isolates were grown to saturation before being diluted to an OD of 0.1 in 3 mL or 1 mL of fresh YP- media, with either 2% glucose or 2% galactose (Sigma), within 48 deep-well plates (Enzyscreen) or BioLector XT/ Microfluidic Flower plates (Beckman Coulter) respectively. Plates then being sealed using Aeraseal film (Sigma) and placed to incubate at 30 °C, with 700 rpm shaking within Multitron Standard incubators (Infors HT) for deep-well plates, or with 800 rpm shaking within Bio-Lector XT Microbioreactor fitted with microfluidic module, for Flower plates. When specified, culture pH was monitored every 10 min, with 1 M NaOH or 1 M HCl automatically supplemented to maintain a constant culture pH of 5.0. After 72 h of cultivation, 10 μL was diluted in 200 μL PBS for analysis of pHluorin2 signal on Attune NxT (VL2-H:BL1-H).

## MitoTracker staining

pHluorin2-expressing lycopene MCF was grown under batch flask conditions for 72 h before 50 μL of culture was sampled, washed twice in PBS, and resuspended in 1 mL PBS. 0.5 μL of 1 M MitoTracker™ Red FM (ThermoFisher) dye was then added, with cells being left to incubate in the dark at 30 °C for 30 min to allow for mitochondrial staining to occur. Post-incubation, cells were washed in PBS to remove any excess dye. 20 μL of washed sample was then diluted in 100 μL PBS and analysed using the Attune NxT flow cytometer, with MitoTracker™ Red FM signal being detected in the YL2-H channel and normalised to FSC-H to account for variations in cell size. 5 μL of cells were also transferred to slide for analysis on Ti2 Twin-Cam-TIRF microscope, with detection of MitoTracker™ Red FM being achieved using Cy5 HQ filter cube.

## Subpopulation-level product quantification

Following 72 h of batched cultivation, cells were diluted 1:50 in 5 mL of PBS and analysed on the CytoFLEX SRT Cell Sorter, with sorting performed using manually defined FSC-A:FSC-H and V525-H:B525-H gates.

Violacein quantification was performed in tandem to pHluorin2 signal acquisition (V525-H:B525-H), through detection of its autofluorescence signal using the R712-H:FSC-H channels – in line with spectral characterisation obtained from Aurora Spectral flow cytometer (Cytek, Supplementary Fig. 19A, B). Signal of 10,000 events were recorded, with clustering of cells based on their pHluorin2 and autofluorescence signals – as detailed in "Subpopulation clustering" section – allowing for quantification of production at the subpopulation-level. Visual confirmation of phenotypes was achieved using fluorescent microscopy. 1.5 x 10$^6$ cells per subpopulation were sorted, pelleted, and resuspended in 10 μL of PBS. 2 μL of resuspended culture was then transferred to a slide and analysed using the Ti2 Twin-Cam-TIRF microscope, with detection of violacein autofluorescence achieved using Cy5 HQ filter cube.

For subpopulation quantification of lycopene, Raman microscopy was used (Supplementary Fig. 19C). 5 x 10$^6$ cells per subpopulation were sorted and fixed through resuspension in 1 mL 70% EtOH. Resuspended cells were incubated at room temperature for 5 min, washed twice in PBS, resuspended in 2 μL of PBS, and plated on a fused silica coverslip (UQG Optics). Cells were then left to dry for a couple of minutes before being covered by a second coverslip, to ensure a dense uniform layer of single cells for analysis. Raman spectra were collected using a system comprised of an inverted optical microscope (DM IRB/ E; Leica) equipped with a 40×/0.85 objective lens (Leica), a 660 nm wavelength laser (Gem, Novanta Photonics), a spectrometer (IsoPlane 320; Princeton Instruments), a back-illuminated deep depletion 1600 × 200 EMCCD (DU970P; Andor Technology), and an automated sample stage (H30XYZ ProScan II; Prior Scientific). A minimum of 300

measurements per subpopulation were collected by raster scanning with a 5 μm step size, as to ensure no cell was measured twice. Pre-processing of Raman spectra involved quality control by visual inspection, smoothing using SG filter, and baseline correction. Quantification of lycopene was achieved by calculating the sum of the area under three lycopene peaks centred at 1,000, 1,148 and 1,508 $cm^{-1}$ with 30 $cm^{-1}$ window on either side, defined using positive control, using integration via trapezoidal method.

## Population-level lycopene quantification

A total of 1 mL of deep-well plate culture was sampled, washed twice in PBS, and resuspended in 250 μL acetone (Sigma). 425–600 μm glass beads (Sigma) were added to resuspended samples before lysing them using the Evolution Tissue Homogeniser (Precellys), with 8 cycles of 60 sec at 8,000 rpm and 15 sec of rest. Samples then pelleted to allow for isolation of lycopene-containing supernatant, which was analysed using HPLC. Another 1 mL of culture was pelleted, its supernatant removed, and its pellet left to dry overnight at 60 °C for dry cell weight (DCW) measurements. Lycopene production to biomass was calculated by taking ratio of lycopene production to DCW.

## Single cell-level lycopene quantification

5 μL of culture was diluted in 900 mL buffer A and 100 mL buffer B (proprietary, Single-Cell Biotech) and processed on yeast microfluidic chip with flow rate of 40 μL/min using Flow-mode Raman-activated Cell Sorter (FlowRACS, Single-Cell Biotech). Approximately 750 cell spectra were collected per condition, with lycopene peaks for lycopene quantification centred at 1,002, 1,154 and 1,513 $cm^{-1}$ with 30 $cm^{-1}$ window on either side, in line with lycopene standard and positive control.

## Heterogeneity detection

Detection of macro-heterogeneity from flow cytometry data was performed as follows. Debris, doublet, and budding yeasts were filtered out by FSC-A:SSC-A and FSC-A:FSC-H gating in FlowJo (BD Life Sciences). Compensation was applied to relevant sensing data using flowCore package[76] in R, with compensation matrix generated from mNeon and mScarletI single-color control analysis on Attune NxT. Heterogeneity within signal distributions was quantified using Hartigans' dip test statistic, as calculated using the R diptest package[77]. Hartigan's dip test works by assessing the likelihood that a given distribution reflects more than one mode. The null hypothesis is that of a uniform distribution, reflective of homogeneity, while deviations from the null hypothesis are quantified through increasing dip test statistics ($D_n$), with high $D_n$ implying distinct subpopulations.

## Subpopulation clustering

When applicable, clustering into subpopulations was performed by fitting two Gaussian distributions of variable shape, volume, and orientation onto multivariate data using R's mClust[78], with best suited model being evaluated and selected based on the Bayesian Information Criterion (BIC). Z-score standardisation of the signals was performed prior to clustering to ensure equal weighing of each clustering variable. Clustering was primarily performed to extract relative p1 frequency values.

## Statistical analysis and reproducibility

All flow cytometry experiments were performed with at least three biological replicates, while characterisation experiments included a minimum of two biological replicates, with the number of technical replicates varying depending on experimental protocol. Visualisation of phenotypes using microscopy was usually performed only once, as a validation of the trends observed using flow cytometry. All attempts at replication were successful, unless specified otherwise. Statistical analysis was performed using the t.test function or wilcox.test test in R, depending on assessed normality.

## Reporting summary

Further information on research design is available in the Nature Portfolio Reporting Summary linked to this article.

## Data availability

Data supporting the findings of this work are available within the paper and its Supplementary Information files. A reporting summary for this Article is available as a Supplementary Information file. Source data are provided with this paper.

## Code availability

All scripts used for data analysis and plotting are available on GitHub [https://github.com/js6015/2025-MetabolicSubpopulations-DataAnalysis].

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

## Acknowledgements

R.L-A. received funding from BBSRC (BB/R01602X/1, BB/T013176/1, BB/T011408/1 - 19-ERACoBioTech- 33 SyCoLim, BB/X01911X/1, BB/Y008510/1 – Engineering Biology Hub for Microbial Foods), EPSRC (AI-4-EB BB/W013770/1, and EEBio Programme Grant EP/Y014073/1), Yeast4Bio Cost Action 18229, European Research Council (ERC) (DEUSBIO - 949080), the Bio-based Industries Joint (PERFECOAT- 101022370) under the European Union's Horizon 2020 research and innovation programme and the European Innovation Council (EIC) under grant agreement No. 101098826 (SKINDEV). Imperial College London UKRI Impact Acceleration Account (EPSRC –EP/X52556X/1, BBSRC -BB/X511055/1). Thanks to the Bezos Earth Fund through the Bezos Centre for Sustainable Protein (BCSP/IC/001). We thank D. Patel for constructing the lycopene-producing parental strain used in this study; T. Ellis for access to the yeast modular cloning toolkit (YTK), including L0 parts containing SCFP3A, mGFPmut2, sfGFP, mVenusNB, mRuby2, and mScarletI, and providing us with the violacein-production plasmid used in the study; J. Fu and S. Tapia Leiva for the analytical HPLC within lab; M. Mülleder and M. Ralser labs for providing the yeast prototrophic toolkit pHLUM v.2; M. Gorwa-Grauslund for providing us with their CEN.PK2-1C *gpd1::TRP1 gpd2::LEU2 his3::HIS3* strain for validation of pGPD2 sensing unit's functionality; and the QIBEBT and Single Cell Biotech for providing us with FlowRACS facilities used for single-cell analysis of lycopene production dynamics.

## Author contributions

J.S., K.S., and R.L-A conceptualised the study. J.S. performed the experimental studies, data analysis, and wrote the manuscript draft. CK aided with fluorescent microscopy studies, Y.S. with Raman microscopy, and P.H. with BioLector runs. M.P. cloned ATP-sensing constructs. M.S. provided access to the London Biofoundry and C.R. to the Raman microscope. R.L-A. supervised the work and obtained funding. All authors contributed and reviewed the final manuscript.

## Competing interests

The authors declare no competing interests.
