## [Peer Review file · Nature Communications]

Single cell profiling framework reveals metabolic subpopulations as drivers of bioproduction heterogeneity

Corresponding Author: Professor Rodrigo Ledesma-Amaro

Version 0:

Reviewer comments:

Reviewer #1

(Remarks to the Author)

This manuscript presents an innovative biosensor-based toolkit integrated with flow cytometry and clustering algorithms to investigate phenotypic heterogeneity in engineered yeast. By leveraging this platform, the authors dissected pH-based subpopulations and metabolite production states, offering a modular strategy to characterize cell-to-cell variation in microbial cell factories. The study is useful in synthetic biology and bioprocess optimization. However, several critical aspects require further validation, deeper exploration, and broader contextualization to enhance the study's impact and reproducibility. The following comments should be addressed before it can be considered for publication.

1. The authors suggest that the identified subpopulation system is suitable for industrial biochemical production. Have the authors tested this approach in scaled-up fermentations for lycopene or violacein production under optimized conditions? Data from larger-scale bioreactor experiments would strengthen the claim of industrial applicability.
2. The subpopulation dynamics were analyzed at 4 and 8 hours. Have the authors investigated longer-term dynamics (e.g., 24 or 48 hours)? Time-resolved data over extended culture durations could reveal population stability, transitions, or emergence of new subpopulations relevant for bioproduction.
3. The study reports differential production of lycopene and violacein between the p1 and p2 subpopulations. Have the authors performed untargeted or targeted metabolite profiling to identify other metabolic differences between these subpopulations? This could help elucidate the underlying metabolic shifts and regulatory networks.
4. The authors should consider incorporating continuous or time-resolved tracking of subpopulation dynamics using microfluidic platforms or time-lapse microscopy. Such approaches could provide finer resolution of how individual cells transition between states, improving mechanistic insights.
5. Since pH is a controllable parameter in bioreactor systems or flask fermentations using buffer systems, have the authors tested how buffering influences biosensor performance and subpopulation behavior? It would be useful to assess how pH control affects metabolite yields and population distribution.
6. The authors compared lycopene production under glucose and galactose conditions. Was a similar comparison performed for violacein production? This would help determine whether the observed metabolic subpopulations are carbon source-dependent across different production modules.
7. Have the authors quantified violacein production across different subpopulations? Quantitative data would allow for a more precise correlation with biosensor outputs and strengthen the conclusions regarding violacein-linked metabolic heterogeneity.
8. Given the importance of redox cofactors like NADH and NADPH in biochemical production, have the authors considered engineering or characterizing subpopulations based on redox biosensors? Comparing NADH- or NADPH-linked subpopulations under different carbon sources (glucose vs. galactose) could uncover additional layers of heterogeneity.

impacting productivity.

9. The authors should include a clearer and more quantitative comparison between the presented biosensor toolkit and existing platforms such as metabolomics or transcriptomics. They should also highlight the advantages in terms of temporal resolution, throughput, sensitivity, and cost to emphasize the novelty and utility of their approach.

10. The authors should provide more detailed validation of each biosensor used in the study. Specifically, please include dose-response curves, dynamic range, linearity, response time, and potential cross-reactivity. This information is critical to assess biosensor robustness and reproducibility.

Reviewer #2

(Remarks to the Author)

In this study, the authors used biological sensors to detect a range of intracellular conditions in yeast at the single-cell level, aiming to characterize heterogeneity among the cell population with a homogenous genetic background that is triggered by stress. This work demonstrates the use of genetic sensors to differentiate cells in different biological states, which has the potential to identify intracellular conditions that facilitate specific cellular activities, such as biosynthesis. The manuscript provides a clear and detailed description of the characterization of genetic parts, which supports the synthetic biology community in using these genetic tools. Here are several concerns and suggestions:

1. In Figure 3, some stresses in this study, such as nutrient depletion, may affect gene expression activities (transcription and/or translation). As genetic sensors involve gene expression, it is critical to validate that the expression rates of sensors and reporters are not affected by the changing conditions. One common way of validating a genetic sensor is to replace the genetic sensor gene with a fluorescent protein gene and show that the expression of that fluorescent protein is not affected by the changing condition.

2. Based on the results in Figure 4, the authors hypothesize that intracellular pH affects the biosynthesis of violacein and lycopene. To validate this hypothesis, they changed the carbon source to galactose to alter cellular pH. However, a change in carbon source also altered metabolic flux, which may affect the activity of biosynthetic pathways. As demonstrated in Figure 2, intracellular pH can be modulated by a supply of a weak acid. The authors should perform a new set of experiments, providing an acid, base, or buffer to adjust the cellular pH for improving biosynthesis yield, which will generate direct, strong support for their hypothesis and show the utility of their single-cell level genetic sensing approach.

3. There are some typos and missing characters throughout the manuscript. For example, in the Figure 2 legend, "(B) Change in pYRE-mNeon signal following exposure to varying concentrations of diamide dashed line)" is missing an open bracket: "(B) Change in pYRE-mNeon signal following exposure to varying concentrations of diamide (dashed line)". In Figure 3A, an x-axis marker is missing.

Reviewer #3

(Remarks to the Author)

This is a very solid paper addressing (1) a framework about how to uncover metabolic heterogeneity, (2) finding a link between heterogeneity in pH and bioproduction, and (3) how to use that information to improve bioproduction. The first part is general and is based largely on the testing of a standardized set of intracellular reporters and associated fluorescent proteins. The second and third parts are a specific exemplar case whereby low glucose conditions trigger a split into distinctly different metabolic phenotypes that affect, in opposite ways, production of lycopene and violacein.

This question is rather important and is only just beginning to build traction in the research community. While it is applied here to bioproduction in yeast, single-cell variability has been seen to be critical to many areas of microbiology, ranging from pathogenesis to antibiotic treatment. This paper excels in terms of providing a useful toolkit for yeast and in providing a key example to point to for why this matters. I have often found people can be convinced heterogeneity exists (it does!) but are less inclined to believe it actually matters for a phenotype of their interest. There we often do not know and have limited examples to use that can come across as anecdotes. This paper is important in providing a pathway to more systematically rooting those out. For both the work on the tools and the pH connection to production of these pigments, the work well-supports the claims.

One suggestion I have would be to be a bit more clear in the writing of the paper of this duality between a general set of tools and approach and then transition to an exemplar that was enabled because several internal states were followed across several conditions to then explore what effect that might have on bioproduction. Having read the title, I was a bit surprised by the flow that ended up occurring and making this more explicit could be useful.

My one concern with the data is that the rapid re-establishment of two phenotypes does not necessarily preclude a genetic basis. Normal point mutations would not have the potential to do this, but hyper-variable genetics can. This could be either regions that flip (phase variation, like what is used with Flp-FRT) or copy-number variations of local duplications (which can be rapidly selected upon and either be as large as the size of gene(s) or as small as single nucleotides). This can be easily addressed: send your populations for sequencing IMMEDIATELY after sorting and use an appropriate analysis pipeline that

can detect these “other” events (like breseq). Pay particular attention to novel junctions or coverage along the genome. Even if you do find a change in this manner, the phenomenon is the same and it is just as important to identify. That said, it would have a different basis (genetic) and could be addressed differently (change the genome to prevent flipping or try to constrain repeat expansion/contraction).

Version 1:

Reviewer comments:

Reviewer #1

(Remarks to the Author)

The authors have addressed my previous comments. However, some aspects still need to be elaborated, and additional details should be provided to improve clarity.

1. The authors should include the micro-bioreactor figure in the main text.
2. Have the authors analyzed lycopene production in deep-well plates, BioLector, and BioLector under buffering conditions? The authors should provide lycopene production data under all three conditions.

Reviewer #2

(Remarks to the Author)

The revised manuscript has improved in clarity with supports from new experiments. All my comments are addressed.

Reviewer #3

(Remarks to the Author)

The responses to my comments (as well as the other reviewers) have addressed my concerns regarding both the title/selling point of the paper and to the concern that genetic changes could be involved. While the latter could still be occurring, I think now that it is recognized it is sufficiently dealt with and can represent further work in the future.

Version 2:

Reviewer comments:

Reviewer #1

(Remarks to the Author)

The authors have addressed my comments.

Point-by-point Response to Reviewers

Reviewer #1 (Remarks to the Author):

This manuscript presents an innovative biosensor-based toolkit integrated with flow cytometry and clustering algorithms to investigate phenotypic heterogeneity in engineered yeast. By leveraging this platform, the authors dissected pH-based subpopulations and metabolite production states, offering a modular strategy to characterize cell-to-cell variation in microbial cell factories. The study is useful in synthetic biology and bioprocess optimization. However, several critical aspects require further validation, deeper exploration, and broader contextualization to enhance the study's impact and reproducibility. The following comments should be addressed before it can be considered for publication.

We would like to thank reviewer 1 for their thorough and insightful comments.

1. The authors suggest that the identified subpopulation system is suitable for industrial biochemical production. Have the authors tested this approach in scaled-up fermentations for lycopene or violacein production under optimized conditions? Data from larger-scale bioreactor experiments would strengthen the claim of industrial applicability.

Scale-up is indeed a critical consideration for bioproduction, especially within the context of industrial applicability, and the reviewer is justified in pointing this out as, while heterogeneity is believed to be amplified in large-scale bioreactors due to uneven mixing, little empirical data exists to demonstrate this in practice. Unfortunately, large-scale fermentation experiments were not feasible within our timeframe; but we have now updated our manuscript to include data which tracks subpopulation dynamics within the BioLector micro-bioreactors. Although these remain small-scale, they mimic industrial bioreactor shaking conditions and are thus more representative of industrial conditions, which we believe strengthen this industrial applicability¹. Future works aimed at further investigating the impact of scale-up on heterogeneity using our framework are currently under consideration for future works.

The new text describing the results reads:

This carbon source-dependent pattern persisted within micro-bioreactor cultivation and under pH buffering conditions, confirming robustness of the observed subpopulation dynamics, including to changes in external pH (Fig. Supplementary Figure 18A).

Modified Supplementary Fig. 18A shows:

Supplementary Figure 18 – Further exploration of pH_i subpopulation dynamics on glucose vs galactose. (A) Effect switching carbon source from glucose to galactose had on the pH_i dynamics, as determined by pHluorin2 signal, of lycopene-producing strains following 72h of cultivation within deep-well plate, BioLector micro-bioreactor, and BioLector with buffering at pH

= 5.0 cultivation, across 3 biological replicates. Significance scores denote p-value $p < 0.05$ (*) based on Kruskal-Wallis test. Error bars represent standard error to the mean.

2. The subpopulation dynamics were analysed at 4 and 8 hours. Have the authors investigated longer-term dynamics (e.g., 24 or 48 hours)? Time-resolved data over extended culture durations could reveal population stability, transitions, or emergence of new subpopulations relevant for bioproduction.

We have explored long-term subpopulation dynamics under batch flask fermentation following 24h, 48h, and 72h of cultivation, with the results displayed in Supplementary Fig. 14. The subpopulations remained stable in their frequency throughout the cultivation period, with p1 representing approximately 20% and 40% of the whole population within the violacein and lycopene microbial cell factories respectively, in accordance with what was observed following shift to CSTR/AVE media.

Text reads:

Fully differentiated subpopulations appeared within both microbial cell factories following 24 hours of flask batch fermentation and persisted throughout the cultivation process (Fig. 4D, Supplementary Fig. 14).

...

p1's signal remained steady upon differentiation, regardless of the product or sampling time, whereas p2's signal showed a gradual decline, dropping from 1.23 and 1.29 to 1.00 and 1.06 a.u. following 24h to 72h of cultivation within violacein and lycopene MCFs respectively (Supplementary Fig. 14). Curiously, while the relative frequency of p1 when compared to p2 remained stable throughout the cultivation process, it differed between the products, enduring at approximately ~20% and ~40% of the total violacein and lycopene producing population respectively.

Supplementary Fig. 14 shows:

Supplementary Figure 14 – Evolution of pH_i dynamics within lycopene and violacein bioproduction. Cell co-transformed with pHLuorin2 and either the lycopene- or violacein-production pathway were grown in batched vs fed-batched-like, characterised by glucose pulses, YPD flask cultures, with their pHLuorin2 signal being recorded and analysed every 24h using flow cytometry.

3. The study reports differential production of lycopene and violacein between the p1 and p2 subpopulations. Have the authors performed untargeted or targeted metabolite profiling to identify other metabolic differences between these subpopulations? This could help elucidate the underlying metabolic shifts and regulatory networks.

We concur with the reviewer's feedback that a comprehensive metabolic profile of the subpopulations is crucial to gain a better understanding of the differentiation mechanism. Traditional untargeted or targeted metabolite profiling, however, require a considerable amount of biomass which can be quite challenging to acquire within the study of subpopulations as we are constrained by the frequency of the less abundant subpopulation, the sorting rate of the FACS we have at our disposal, and the stability of pHluorin2 signal under prolonged non-experimental conditions required for FACS sorting. Performing metabolic profiling is thus not straight forward and requires further methodological or equipment development, something that we are exploring for future works.

4. The authors should consider incorporating continuous or time-resolved tracking of subpopulation dynamics using microfluidic platforms or time-lapse microscopy. Such approaches could provide finer resolution of how individual cells transition between states, improving mechanistic insights.

We agree with the reviewer and have now updated our manuscript to include new time-lapse microscopy data which analyses the pH_i of single cells in response to shift to CSTRAVE media. While we were unable to track pH_i of cells pre-shift due to the high autofluorescence of YPD media, we did obtain robust data showing pH_i dynamics immediately post shift. In doing so, we observed all cells to experience a slight drop in pH_i post shift, with the primary difference between cells from the two subpopulations seemingly being their ability to counteract this drop. Indeed, cells which ended up in p2 had their pH_i bounce back up shortly post being shifted while those which ended up in p1 kept on experiencing a gradual decline in pH_i which eventually stabilised 4h post shift.

Our time-lapse microscopy results also showed the subpopulations to remain stable over time, with no cells transitioning between subpopulations during our 15h observation period. As cells do not divide under our shift condition, we were unable to confirm whether subpopulation's phenotype is inherited by daughter cells, but we did anecdotally observe bud cells which had been forming pre-shift to display the same phenotype as mother cells, suggesting that it may indeed be an inheritable phenotype – in line with the observations from Bagamery et al².

New text reads:

*A closer look at differentiation dynamics using time-lapse microscopy revealed all cells to experience a slight drop in pH_i immediately post shift, with the pH_i of p2 cells bouncing back as time progressed while that of p1 cells kept dropping, until it stabilised at around 4h post shift (**Fig. 3D, E, Supplementary Fig. 13A**). These results also showed the subpopulations to remain stable throughout the observation period, persisting 15h post shift with no cells transitioning between the subpopulations.*

Fig. 3D and E and Supplementary Fig. 13A show:

Figure 3 – Metabolic response to environmental perturbations. (D) Tracked changes in pHluorin2 signal of single cells post shift to CSTRAVE media (left). Tracked signals normalised to initial pH_i (t = 0h) to reduce noise and allow for inter-cell comparison of pH_i dynamics, with raw endpoint signals (t = 15h) for p1 and p2 subpopulations shown (right).

Supplementary Figure 13 – Further characterisation of pH_i subpopulations in response to CSTRAVE shift. (A) Timelapse microscopy of pHluorin2-expressing cells post shift to CSTRAVE media. pHluorin2 signal was determined by comparing cell's relative fluorescence following 375nm and 480nm excitation, with increased signal following 375nm excitation compared to 480nm excitation indicative of greater pHluorin2 values.

5. Since pH is a controllable parameter in bioreactor systems or flask fermentations using buffer systems, have the authors tested how buffering influences biosensor performance and subpopulation behavior? It would be useful to assess how pH control affects metabolite yields and population distribution.

As part of our added BioLector experiment, discussed in comment (1), we have included a condition in which the pH of the media was buffered (pH = 5.0), in accordance with industry practices. In doing so, we observed that subpopulations dynamics persisted independent of buffering conditions, with buffering having a minimal impact on the relative frequency of p1 across both glucose and galactose cultivation. We believed this demonstrates the robustness of pHluorin2 as a subpopulation marker as well as the robustness of our observed subpopulation dynamics.

6. The authors compared lycopene production under glucose and galactose conditions. Was a similar comparison performed for violacein production? This would help determine whether the observed metabolic subpopulations are carbon source-dependent across different production modules.

We have carried out the experiment suggested by the reviewer, highlighting the effect of glucose vs galactose conditions on pH_i dynamics of violacein-producing strain, which can be found in

Supplementary Fig. 18B. Populations grown on galactose exhibited a 22% reduction in p1 frequency compared to glucose-grown cells, consistent with trends observed in lycopene-producing strains, albeit less pronounced potentially as a result of violacein-producing strains already having relatively low baseline p1 frequencies. While a drop was observed, it was found to be not statistically significant, potentially due to the low number of biological replicates ($n = 3$). This reduction in p1 frequencies likely resulted in reduced bioproduction efficiencies, with cells grown on glucose producing on average 0.33 mg.L^{-1} of violacein while cells grown on galactose produced amounts too low to be reliably quantified using HPLC.

New text reads:

Of note, switching carbon source from glucose to galactose had a similar effect of the pH_i dynamics of violacein-producing populations, with its relative p1 frequency dropping 22% from glucose to galactose cultivation (Supplementary Fig. 18B).

Supplementary Fig. 18B shows:

Supplementary Figure 18 – Further exploration of pH_i subpopulation dynamics on glucose vs galactose. (B) Effect switching carbon source from glucose to galactose had on the pH_i dynamics, as determined by pHluorin2 signal, of violacein-producing strains following 72h of BioLector cultivation, across 3 biological replicates. Significance scores denote p-value with $p > 0.05$ (*ns*) based on Kruskal-Wallis test. Error bars represent standard error to the mean.

7. Have the authors quantified violacein production across different subpopulations? Quantitative data would allow for a more precise correlation with biosensor outputs and strengthen the conclusions regarding violacein-linked metabolic heterogeneity.

As mentioned previously, quantification of metabolites, including of violacein, at a subpopulation-level can be challenging due to the biomass required for robust quantification using analytical methods. Instead, we opted to quantify violacein within the subpopulations using autofluorescence, having performed spectral characterisation of violacein-producing and non-producing strains and found that emission between 605nm and 707nm following red laser excitation resulted in the strongest autofluorescence detection signal. We have now added this data to the manuscript as validation of autofluorescence as a method for violacein quantification (see Supplementary Figure 19). As further validation of this methodology, we sampled violacein-producing cultures following 48h, 72h and 120h and compared the resulting population-averaged autofluorescence to violacein production as quantified by HPLC, showing a positive correlation between the two.

New methods reads:

Single-cell violacein autofluorescence quantification was performed in tandem to pHluorin2 signal quantification, using the R712-H:FSC-H, in line with spectral characterisation obtained from Aurora Spectral flow cytometer (Cytex, **Supplementary Fig. 19A, B**).

Supplementary Figure 19 and Figure R1 show:

Supplementary Figure 19 – Production quantification controls. (A) Spectral characterisation following excitation with yellow-green (YG) or red (R) lasers of strains co-expressing pHluorin2 within violacein production pathway compared to pHluorin2-expressing and pHLUM control strains. Channels 1 through 10 represent different emission properties, with R3-A, corresponding to an emission wavelength between 688nm and 707nm, displaying the strongest signal in response to violacein expression. (B) Quantification of violacein autofluorescence within strains expressing the violacein production pathway when compared to pHLUM negative control strains using red laser excitation within Cytotex.

Figure R1 – Validation of autofluorescence for violacein quantification. Correlation between violacein quantification as determined using autofluorescence vs extraction and HPLC.

8. Given the importance of redox cofactors like NADH and NADPH in biochemical production, have the authors considered engineering or characterizing subpopulations based on redox

biosensors? Comparing NADH- or NADPH-linked subpopulations under different carbon sources (glucose vs. galactose) could uncover additional layers of heterogeneity impacting productivity.

Following the suggestion from the reviewer, we analysed NADH and NADPH dynamics using the pGPD2-mNeon and YRE-mNeon sensing signal respectively. Results showed a homogeneous distribution of cofactors upon both glucose and galactose cultivation. The lack of distinguishable subpopulations does not, however, rule out a potential impact of redox co-factors and future works can look deeper into this.

Figure 2R shows:

Figure R2 – NADH and NADPH subpopulation dynamics under glucose vs galactose batch flask fermentation. pGPD2-mNeon and YRE-mNeon expressing cells were cultivated within batch flask fermentation using glucose vs galactose as a carbon source, with the resulting sensing signal distributions assayed following 24h, 48h and 72h of cultivation and analysed using flow cytometry.

9. The authors should include a clearer and more quantitative comparison between the presented biosensor toolkit and existing platforms such as metabolomics or transcriptomics. They should also highlight the advantages in terms of temporal resolution, throughput, sensitivity, and cost to emphasize the novelty and utility of their approach.

We appreciate this suggestion and have now edited the introduction to better emphasize the advantages of genetically encoded biosensors. Of note, genetically encoded biosensors and traditional metabolomics tend to be seen as complementary rather than alternatives and, as such, quantitative comparison of the two is not prevalent within the literature. Both have their own targets, strengths and weaknesses which makes them more or less suited for certain uses.

New text reads:

Quantification of metabolites at the single-cell level, however, is technically challenging due to the low sensitivity of existing metabolomics platforms. While efforts to address this are underway, single-cell metabolomics remain in their infancy and are thus prohibitively costly, low throughput, and not readily accessible¹⁹.

10. The authors should provide more detailed validation of each biosensor used in the study. Specifically, please include dose-response curves, dynamic range, linearity, response time, and potential cross-reactivity. This information is critical to assess biosensor robustness and reproducibility.

We agree that comprehensive characterisation of each biosensor would be valuable. Here, we employed biosensors that have been previously characterised within the literature and have provided clear references to the original publications for readers seeking further validation data. A comprehensive characterisation of biosensors presents significant technical challenges, necessitating a mix of *in vitro* and cell-free assays which require specialised expertise and extensive assay optimisation. Indeed, such data can often be omitted from publications, even those presenting novel biosensors, due to the substantial technical barriers posed, especially when targeting metabolites as complex and fiddly as core intracellular metabolites. Therefore, we believe that additional characterisation falls outside the scope of our current study.

In this work, we primarily aimed to showcase the value of previously characterised biosensors to detect and study isogenic heterogeneity. Upon publication of our work, will make our biosensors cassettes and strains available to anyone in the community who request them to perform further characterisation.

Reviewer #2 (Remarks to the Author):

In this study, the authors used biological sensors to detect a range of intracellular conditions in yeast at the single-cell level, aiming to characterize heterogeneity among the cell population with a homogenous genetic background that is triggered by stress. This work demonstrates the use of genetic sensors to differentiate cells in different biological states, which has the potential to identify intracellular conditions that facilitate specific cellular activities, such as biosynthesis. The manuscript provides a clear and detailed description of the characterization of genetic parts, which supports the synthetic biology community in using these genetic tools. Here are several concerns and suggestions:

We thank reviewer 2 for their valuable comments and have now addressed these as described below, improving the manuscript accordingly.

1. In Figure 3, some stresses in this study, such as nutrient depletion, may affect gene expression activities (transcription and/or translation). As genetic sensors involve gene expression, it is critical to validate that the expression rates of sensors and reporters are not affected by the changing conditions. One common way of validating a genetic sensor is to replace the genetic sensor gene with a fluorescent protein gene and show that the expression of that fluorescent protein is not affected by the changing condition.

Thanks for the comment. Three different types of biosensors were used in this study, including transcription factor-based, ribozyme-based and FRET-based biosensors. FRET-based biosensors are ratiometric by design, measuring the fluorescence ratio following excitation or emission at distinct wavelengths, and thus, automatically taking into account any variation in expression levels. For our transcription factor-based and ribozyme-based biosensors, we followed the strategy mentioned by the reviewer, placing a control fluorescent protein under constitutive expression immediately downstream of the biosensor, with the resulting measured signal being

the fluorescence of the sensing reporter normalised to that of the control fluorescent protein, and thereby also accounting for any variation in expression level. In the case of our ribozyme-based biosensor, the constitutive promoters used to express the sensing construct and control fluorescent protein were pTEF1 and pTEF2 respectively, two promoters which have comparable and correlated expression levels³.

2. Based on the results in Figure 4, the authors hypothesize that intracellular pH affects the biosynthesis of violacein and lycopene. To validate this hypothesis, they changed the carbon source to galactose to alter cellular pH. However, a change in carbon source also altered metabolic flux, which may affect the activity of biosynthetic pathways. As demonstrated in Figure 2, intracellular pH can be modulated by a supply of a weak acid. The authors should perform a new set of experiments, providing an acid, base, or buffer to adjust the cellular pH for improving biosynthesis yield, which will generate direct, strong support for their hypothesis and show the utility of their single-cell level genetic sensing approach.

The reviewer raises a thoughtful point, and we concur that the correlation we observe between improved bioproduction and altered subpopulations dynamics may also arise from factors independent of subpopulations dynamics, such as metabolic rewiring brought by the use of an alternative carbon source. The reviewer also makes insightful proposal to counter this bias, by using culture pH as a method of modulating subpopulation dynamics instead. To test this, we leveraged the BioLector mini bioreactor system and assayed the impact buffering of culture pH would have on subpopulation dynamics of lycopene-producing strains. Culture pH was constantly monitored, being buffered with either HCl or NaOH to ensure a pH of 5.0 was maintained. In doing so, we observed robust subpopulation differentiation independent of buffering conditions, with pH buffering having a minimal impact on p1 frequency across both glucose and galactose conditions. As suggested by the reviewer, we also tested the addition of 10mM acetic acid which did not impact subpopulation dynamics, further suggesting that the observed differentiation pattern is robust to, and thus unfortunately cannot be modulated, through culture pH.

While it may be impossible to fully detangle metabolic rewiring from subpopulation dynamics, we have now included single-cell lycopene production data to assess the differences in production dynamics between glucose and galactose cultivation. In doing so, we show galactose cultivation to induce a shift from low- to higher- producing phenotypes when compared to glucose cultivation, with this shift paralleling that observed within the pH_i dynamics. We believe this lends further credibility to the hypothesis that the increase in production efficiency under galactose cultivation is, at least in part, driven by modulation of subpopulation dynamics.

New text reads:

Results showed the relative frequency of p1 to drop significantly under galactose conditions, with p1 comprising 70.6% of the total cell population when grown on glucose and dropping to 29.8% under galactose cultivation conditions (Fig. 5A, Supplementary Figure 18A). This altered subpopulation dynamics paralleled the changes observed in single-cell lycopene production under both culture conditions, whereby galactose cultivation prompted a shift from low- to high-producing phenotypes when compared to glucose cultivation (Fig. 5B).

...

This carbon source-dependent pattern persisted within micro-bioreactor cultivation and under pH buffering conditions, confirming robustness of the observed subpopulation dynamics, including to changes in external pH (Fig. Supplementary Figure 18A).

Fig. 5AB, Supplementary Fig. 18A, Fig. 3R show:

Figure 5 – Modulation of subpopulation dynamics through cultivation strategies. (A) Effect switching carbon source from glucose to galactose had on the pH_i dynamics, as determined by pHluorin2 signal, of lycopene-producing strains following 72h of cultivation. Clustering of cells into p1 and p2 subpopulations was performed according to Methods, with the relative frequency of p1 displayed. (B) Change in single-cell lycopene production dynamics, as determined by Raman peak AUC, following 72h of cultivation on glucose vs galactose.

Supplementary Figure 18 – Further exploration of pH_i subpopulation dynamics on glucose vs galactose. (A) Effect switching carbon source from glucose to galactose had on the pH_i dynamics, as determined by pHluorin2 signal, of lycopene-producing strains following 72h of cultivation within deep-well plate, BioLector micro-bioreactor, and BioLector with buffering at pH = 5.0 cultivation, across 3 biological replicates. Significance scores denote p-value $p < 0.05$ (*) based on Kruskal-Wallis test. Error bars represent standard error to the mean.

Figure 3R – Impact of weak acid supplementation on pH_i dynamics of lycopene-producing strains. Effect 10mM acetic acid supplementation had on the pH_i dynamics of lycopene-producing strains, as determined by pHluorin2 signal, following 72h of cultivation.

3. There are some typos and missing characters throughout the manuscript. For example, in the Figure 2 legend, “(B) Change in pYRE-mNeon signal following exposure to varying concentrations of diamide dashed line)” is missing an open bracket: “(B) Change in pYRE-mNeon signal following exposure to varying concentrations of diamide (dashed line)”. In Figure 3A, an x-axis marker is missing.

We thank the reviewer for bringing this to our attention. We have now reviewed the manuscript and removed all typos to the best of our ability.

Reviewer #3 (Remarks to the Author):

This is a very solid paper addressing (1) a framework about how to uncover metabolic heterogeneity, (2) finding a link between heterogeneity in pH_i and bioproduction, and (3) how to use that information to improve bioproduction. The first part is general and is based largely on the testing of a standardized set of intracellular reporters and associated fluorescent proteins. The second and third parts are a specific exemplar case whereby low glucose conditions trigger a split into distinctly different metabolic phenotypes that affect, in opposite ways, production of lycopene and violacein.

This question is rather important and is only just beginning to build traction in the research community. While it is applied here to bioproduction in yeast, single-cell variability has been seen to be critical to many areas of microbiology, ranging from pathogenesis to antibiotic treatment. This paper excels in terms of providing a useful toolkit for yeast and in providing a key example to point to for why this matters. I have often found people can be convinced heterogeneity exists (it does!) but are less inclined to believe it actually matters for a phenotype of their interest. There we often do not know and have limited examples to use that can come across as anecdotes. This paper is important in providing a pathway to more systematically rooting those out. For both the work on the tools and the pH connection to production of these pigments, the work well-supports the claims.

We would like to thank Reviewer 3 for sharing our enthusiasm with regards to our results and highlighting the importance of this work.

One suggestion I have would be to be a bit more clear in the writing of the paper of this duality between a general set of tools and approach and then transition to an exemplar that was enabled because several internal states were followed across several conditions to then explore what

effect that might have on bioproduction. Having read the title, I was a bit surprised by the flow that ended up occurring and making this more explicit could be useful.

We thank the reviewer for pointing this out and have now edited the title as well as the abstract and introduction in the hopes of better reflecting the flow of the paper. The new title now reads: “*Metabolic subpopulations as drivers of bioproduction heterogeneity: From framework development to bioprocess optimisation*”. To emphasise the tool development part of the work a bit further, we have now named the sensor toolkit: YTK-ScBiosense and referred to it throughout the document.

My one concern with the data is that the rapid re-establishment of two phenotypes does not necessarily preclude a genetic basis. Normal point mutations would not have the potential to do this, but hyper-variable genetics can. This could be either regions that flip (phase variation, like what is used with Flp-FRT) or copy-number variations of local duplications (which can be rapidly selected upon and either be as large as the size of gene(s) or as small as single nucleotides). This can be easily addressed: send your populations for sequencing IMMEDIATELY after sorting and use an appropriate analysis pipeline that can detect these “other” events (like breseq). Pay particular attention to novel junctions or coverage along the genome. Even if you do find a change in this manner, the phenomenon is the same and it is just as important to identify. That said, it would have a different basis (genetic) and could be addressed differently (change the genome to prevent flipping or try to constrain repeat expansion/contraction).

We appreciate the reviewer's feedback on this point, which raises an important and legitimate concern. Addressing it through sequencing, however, is challenging due to the number of cells required for adequate coverage, our sorting rate being constrained by the frequency of the p1 subpopulation and the limitations of the FACS we have at our disposal, and the stability of pHluorin2 signal under prolonged sorting conditions. Instead, we have opted to use an alternative strategy to account for the potential impact of mutations, as was done previously by Shabestary et al.⁴, which consists of sorting subpopulations post shift based on pH_i and plating them on rich YPD media to reset their phenotype. We then selected three separate colony isolates from both plated subpopulations, shifted them back to CSTRIVE media and, in doing so, showed both subpopulation phenotype to emerge irrespective of the isolates' history, i.e. of whether they belonged to p1 or p2 in the initial sort. This suggests that the phenotype is indeed isogenic, as, if the cause was mutational, we would expect the subpopulation's phenotype to have persisted upon repeat shift, with cells from p2 only giving rise to p2 phenotype and cells from p1 only giving rise to the p1 phenotype. While these results were initially included within our Supplementary Figures, this reviewer's comment has prompted us to highlight it more within the manuscript, moving the plot to Fig. 3 and emphasizing its significance within the main text. Of note, we recognise that, while our current method is commonly accepted within the literature, it does not fully eliminate the potential impact of reversible genetics events and thus, future works can target the elucidation of such potential effects. We have now included a note in our conclusion to caveat for the potential of such an occurrence.

New text reads:

Lastly, the sorted subpopulations were plated on rich YPD media to reset their phenotype before being shifted back to CSTRIVE media. Doing so resulted in the emergence of p1 and p2 within all of the colony isolates tested, irrespective of the subpopulation's history, thereby confirming the subpopulations to occur as a result of isogenic heterogeneity and not recurring spontaneous mutations (Fig. 3F).

...

Future multi-omics works aimed at elucidating the documented divergent product-dependent responses and their underlying mechanisms – including to account for potential reversible genetic mutations – could not only inform additional cultivation strategies but also engineering targets to enhance both strain productivity and stability.

Figure 3E shows:

Figure 3 – Metabolic response to environmental perturbations. (E) pHluorin2-expressing cells were sorted into p1 vs p2 subpopulations 4h post shift to CSTRAVE media and plated on fresh YPD plates to reset phenotype. 3 separate colony isolates from each plated subpopulation were then shifted back to CSTRAVE media, with the resulting pHluorin2 signal distribution displayed. Assignment of cells to p1 vs p2 subpopulation performed according to Methods.

References:

1. Fink, M., Cserjan-Puschmann, M., Reinisch, D. & Striedner, G. High-throughput microbioreactor provides a capable tool for early stage bioprocess development. *Sci Rep* **11**, 2056 (2021).
2. Bagamery, L. E., Justman, Q. A., Garner, E. C. & Murray, A. W. A Putative Bet-Hedging Strategy Buffers Budding Yeast against Environmental Instability. *Curr Biol* **30**, 4563-4578.e4 (2020).
3. Peng, B., Williams, T. C., Henry, M., Nielsen, L. K. & Vickers, C. E. Controlling heterologous gene expression in yeast cell factories on different carbon substrates and across the diauxic shift: a comparison of yeast promoter activities. *Microb Cell Fact* **14**, 91 (2015).
4. Shabestary, K. *et al.* Phenotypic heterogeneity follows a growth-viability tradeoff in response to amino acid identity. *Nat Commun* **15**, 6515 (2024).

Point-by-point Response to Reviewers

Reviewer #1 (Remarks to the Author):

The authors have addressed my previous comments. However, some aspects still need to be elaborated, and additional details should be provided to improve clarity.

1. The authors should include the micro-bioreactor figure in the main text.

We have now added the micro-bioreactor data to the main text figure.

See modified Figure 5E:

Figure 5 – Modulation of subpopulation dynamics through cultivation strategies. (A) Effect switching carbon source from glucose to galactose had on the pH_i dynamics, as determined by pHluorin2 signal, of lycopene-producing strains following 72h of cultivation. Clustering of cells into p1 and p2 subpopulations was performed according to Methods, with the relative frequency of p1 displayed. Resulting change in (B) single-cell lycopene production dynamics, as determined by Raman peak AUC, (C) lycopene production, as quantified using HPLC, and (D) lycopene production relative to biomass. Dry cell weight (DCW) used as a measure of biomass. (E) Carbon source-dependent shift in subpopulation dynamics conserved across 3 biological replicates within deep-well plate, BioLector micro-bioreactor, and BioLector with buffering at pH = 5.0 cultivation. Significance scores denote *p*-value with *p* < 0.05 (*) based Kruskal-Wallis test. Error bars represent standard error to the mean.

2. Have the authors analysed lycopene production in deep-well plates, BioLector, and BioLector under buffering conditions? The authors should provide lycopene production data under all three conditions.

As suggested by the reviewer, we have now included this data and referenced it within the main text. These experiments showed the expected trend, although the biolector data presented larger variability compared to the previous results obtained in deep well plates. This variability is likely due to the smaller volumes that can be run in the biolector, which only

generate 5-10 mg DCW, while deep well plate cultivation produces over 50 mg DCW. The smaller amount of DCW amplifies errors during the extraction and quantification of lycopene, which is attributed to technical challenges, including the difficulty in achieving homogeneous pigment extraction, low pigment stability upon extraction, and less accurate dry cell weight (DCW) measurements. These factors contribute to the observed dispersion of the data. Future work to study in more detail the effect of pH control on subpopulations and production should consider a larger number of replicates and optimise the extraction protocol for smaller volumes. Due to limitations in accessing the equipment, we have been unable to perform additional experiments on the biolector.

New text reads:

Trends in production under these conditions were also consistent with those observed within deep-well plates, although the reduced culture volumes and biomass of micro-bioreactors promoted variability within the resulting production measurements (Supplementary Fig. 18A).

See modified Supplementary Fig. 18:

Supplementary Figure 18 – Further exploration of pH; subpopulation dynamics following glucose vs galactose cultivation. (A) Effect switching carbon source from glucose to galactose had on the lycopene production relative to DCW of lycopene-producing strains following 72h of cultivation within BioLector micro-bioreactor and BioLector with buffering at pH = 5.0 cultivation. (B) Shift in subpopulation dynamics following 72h of BioLector cultivation on glucose vs galactose conserved within violacein-producing strains. Significance scores denote p-value with $p > 0.05$ (*ns*) based on Kruskal-Wallis test. Error bars represent standard error to the mean, across 3 biological replicates.

Reviewer #2 (Remarks to the Author):

The revised manuscript has improved in clarity with supports from new experiments. All my comments are addressed.

We appreciate the reviewer's time and feedback, and are pleased that we could address their concerns to their satisfaction.

Reviewer #3 (Remarks to the Author):

The responses to my comments (as well as the other reviewers) have addressed my concerns regarding both the title/selling point of the paper and to the concern that genetic changes could be involved. While the latter could still be occurring, I think now that it is recognized it is sufficiently dealt with and can represent further work in the future.

We appreciate the reviewer's time and feedback, and are pleased that we could address their concerns to their satisfaction.